# Contribution of carbonatite and recycled oceanic crust to petit-spot lavas on the western Pacific Plate

Kazuto Mikuni[1,2]*, Naoto Hirano[2,3], Shiki Machida[4], Hirochika Sumino[5], Norikatsu Akizawa[6], Akihiro Tamura[7], Tomoaki Morishita[7], Yasuhiro Kato[4,8,9]

[1] AIST, Geological Survey of Japan, Research Institute of Geology and Geoinformation, Central 7, 1-1-1, Higashi, Tsukuba, Ibaraki 305-8567, Japan.

[2] Graduate School of Science, Tohoku University, 6-3 Aramaki-Aoba, Aoba-ku, Sendai 980–8578, Japan.

[3] Center for Northeast Asian Studies, Tohoku University, 41 Kawauchi, Aoba-ku, Sendai 980–8576, Japan.

[4] Ocean Resources Research Center for Next Generation, Chiba Institution of Technology, 2-17-1 Tsudanuma, Narashino 275-0016, Japan.

[5] Research Center for Advanced Science and Technology, the University of Tokyo, 4-6-1 Komaba, Meguro-ku, Tokyo 153-8904, Japan

[6] Atmosphere and Ocean Research Institute, the University of Tokyo, 5-1-5, Kashiwanoha, Kashiwa 277-8564, Japan.

[7] Earth Science Course, Kanazawa University, Kakuma, Kanazawa 920-1192, Japan.

[8] Department of Systems Innovation, School of Engineering, The University of Tokyo, 7-3-1 Hongo, Bunkyo-ku, Tokyo 113-8656, Japan.

[9] Submarine Resources Research Center, Research Institute for Marine Resources Utilization, Japan Agency for Marine-Earth Science and Technology (JAMSTEC), 2-15 Natsushima-cho, Yokosuka, Kanagawa, 237-0061, Japan.

* *Correspondence to* Kazuto Mikuni (kazuto.mikuni @aist.go.jp)

Authors' e-mail addresses and ORCiD numbers

| | | |
|---|---|---|
| Kazuto Mikuni[1,2]* | kazuto.mikuni@aist.go.jp | 0000-0001-6939-4333 |
| Naoto Hirano[2,3] | nhirano@tohoku.ac.jp | 0000-0003-0980-3929 |
| Shiki Machida[4] | shiki.machida@p.chibakoudai.jp | 0000-0002-1069-7214 |
| Hirochika Sumino[5] | sumino@igcl.c.u-tokyo.ac.jp | 0000-0002-4689-6231 |
| Norikatsu Akizawa[6] | akizawa@g.ecc.u-tokyo.ac.jp | 0000-0003-4210-1160 |
| Akihiro Tamura[7] | aking826@gmail.com | 0000-0002-9112-7976 |
| Tomoaki Morishita[7] | moripta@gmail.com | 0000-0002-8724-6868 |
| Yasuhiro Kato[4,8,9] | ykato@sys.t.u-tokyo.ac.jp | 0000-0002-5711-8304 |

The manuscript is going to be submitted to *Solid Earth*.

**Keywords: Petit-spot volcano, alkali basalt, carbonatite, asthenosphere**

**Abstract**

Petit-spot volcanoes, occurring due to plate flexure, have been reported globally. As the petit-spot melts ascend from the asthenosphere, they provide crucial information of the lithosphere–asthenosphere boundary. Herein, we examined the lava outcrops of six monogenetic volcanoes formed by petit-spot volcanism in the western Pacific. We then analyzed the $^{40}Ar/^{39}Ar$ ages, major and trace element compositions, and Sr, Nd, and Pb isotopic ratios of the petit-spot basalts. The $^{40}Ar/^{39}Ar$ ages of two monogenetic volcanoes were ca. 2.6 Ma (million years ago) and ca. 0 Ma. The isotopic compositions of the western Pacific petit-spot basalts suggest geochemically similar melting sources. They were likely derived from a mixture of high-$\mu$ (HIMU) mantle-like and enriched mantle (EM)-1-like components related to carbonatitic/carbonated materials and recycled crustal components. The characteristic trace element composition (i.e., Zr, Hf, and Ti depletions) of the western Pacific petit-spot magmas could be explained by the partial melting of ~5% crust-bearing garnet lherzolite with 10% carbonatite flux to a given mass of the source, as implied by a mass balance-based melting model. This result confirms the involvement of carbonatite melt and recycled crust in the source of petit-spot melts. It provides insights into the genesis of tectonic-induced volcanoes, including Hawaiian North Arch and Samoan petit-spot-like rejuvenated volcanoes, that have similar trace element composition to petit-spot basalts.

59

60

**Short Summar**y

62

Plate tectonics theory is the motion of rocky plates (lithosphere) over ductile zones (asthenosphere). The causes of the lithosphere–asthenosphere boundary (LAB) are controversial; however, petit-spot volcanism supports the presence of melt at the LAB. We conducted geochemistry, geochronology, and geochemical modeling of petit-spot volcanoes on the western Pacific Plate, and the results suggested that carbonatite melt and recycled oceanic crust induced the partial melting at the LAB.

69

**1 Introduction**

71

Among the upper mantle-derived alkali basaltic lavas in oceanic settings, those on thicker plates away from the mid-ocean ridge, could be divided into plume-related and non-plume-related volcanoes. Plume-related North Arch and post-erosional (rejuvenated-stage) volcanoes have been reported in Hawaii and Samoa (Bianco et al., 2005; Bizimis et al., 2013; Clague and Frey, 1982; Clague and

Moore, 2002; Dixon et al., 2008; Frey et al., 2000; Garcia et al., 2016; Hart et al., 2004; Konter and
Jackson, 2012; Koppers et al., 2008; Reinhard et al., 2019; Yang et al., 2003). Nonplume-related
intraoceanic alkali volcanoes, known as petit-spot volcanoes, probably originate where nearby plate
subduction causes plate flexures and upwelling of asthenospheric magma (Hirano et al., 2006; Hirano
and Machida, 2022; Machida et al., 2015, 2017; Yamamoto et al., 2014, 2018, 2020). The occurrence
of petit-spot volcanisms supports the presence of melt at the lithosphere–asthenosphere boundary
(LAB) below the area at least.
The occurrence of melt in the uppermost asthenosphere could be attributed to small-scale
convection, the presence of hydrous or carbonatitic components, or the uplift of the lithosphere in
response to plate flexure; however, the possibility of such an occurrence remains ambiguous (e.g.,
Bianco et al., 2005; Hua et al., 2023; Korenaga, 2020). The presence of $CO_2$ and
carbonated/carbonatitic materials is a significant factor in the formation of alkaline, silica-
undersaturated melt in the upper mantle (Dasgupta and Hirschmann, 2006; Dasgupta et al., 2007,
2013; Kiseeva et al., 2013; Novella et al., 2014). Experimental studies have shown that the solidus of
carbonate-bearing peridotite is lower than that of $CO_2$-free peridotite (Falloon and Green, 1989. 1990;
Foley et al., 2009; Ghosh et al., 2009). Moreover, carbonatites and Si-undersaturated melts are
generated through the partial melting of $CO_2$-bearing or carbonated peridotite. The produced melts
can exhibit continuous chemical variations depending on pressure (i.e., depth). Carbonatitic melts are
produced in the deep asthenosphere (300–110 km), while carbonated or alkali silicate melts are
generated in the shallower upper mantle (from ~110 to ~75 or 60 km) (Keshav and Gudfinnsson, 2013;
Massuyeau et al., 2015, 2021). Primary carbonated silicate magma and evolved alkali basalts have
been simultaneously observed at the post-spreading ridge in the South China Sea (Zhang et al., 2017;
Zhong et al., 2021). The occurrence of Hawaiian rejuvenated volcanoes can be attributed to a
carbonatite-metasomatized source with or without silicate metasomatism (Borisova and Tilhac, 2021;
Dixon et al., 2008; Zhang et al., 2022).
Submarine petit-spot volcanoes on the subducting northwestern (NW) Pacific Plate may have
originated from carbonate-bearing materials and crustal components (pyroxenite/eclogite) based on
characteristic trace elements, enriched mantle (EM)-1-like Sr, Nd, and Pb isotopic, and relatively low
Mg isotopic compositions (Liu et al., 2020; Machida et al., 2009, 2015). Particularly, the depletion of
specific high-field-strength elements (HFSEs) (i.e., Zr, Hf, and Ti) and the abundance of $CO_2$ in petit-
spot basalts imply that their melting sources are related to carbonated materials (Hirano and Machida,
2022; Okumura and Hirano, 2013). The nature of the uppermost part of the asthenosphere beneath the
oldest Pacific Plate aged 160 Ma was characterized using the eruptive ages and geochemical properties
of six newly observed petit-spot volcanoes and lava outcrops. We verified the contribution of
carbonatitic components and crustal materials to the melting source of petit-spot volcanoes to
understand the nature of the underlying lithosphere–asthenosphere system and model the geodynamic
evolution of the region.

**2 Background**

Over the last 20 years, there has been an increase in the understanding of petit-spot volcanic
settings, providing valuable insights into the nature of the lithosphere–asthenosphere system,
particularly in the NW Pacific region (Hirano et al., 2006; Hirano and Machida, 2022). As other
implications, subducted petit-spot volcanic fields with geological disturbances on the seafloor play a
role in controlling the hypocentral regions of megathrust earthquakes (Fujiwara et al., 2007; Fujie et
al., 2020; Akizawa et al., 2022). Additionally, the vestige of hydrothermal activity due to petit-spot
magmatism has recently been reported (Azami et al., 2023).
Petit-spot melts emerging from the asthenosphere, which are unrelated to mantle plume, could
play a crucial role in clarifying the nature of the LAB (Hirano and Machida, 2022). Their
asthenospheric origin was supported by MORB-like noble-gas isotopic ratios, multi-phase saturation
experiment, and geochemistry (Hirano et al., 2006; Hirano and Machida, 2022; Machida et al., 2015,
2017; Yamamoto et al., 2018). The LAB is recognized as a discontinuous transition in seismic
velocities at the base of the lithosphere, and its causes are attributed to hydration, melting, and mineral
anisotropy with considerations for the unique characteristics in each tectonic setting (e.g., Rychert and
Shearer, 2009). The occurrence of petit-spot volcanoes confirms the existence of melt at the LAB
beneath the area at least (Hirano et al., 2006). Recently, similar volcanic activities have been observed
globally, including in Java (Sunda) Trench, Tonga Trench, Chile Trench, Mariana Trench, Costa Rica,
North American Basin and Range, and the southern offshore of Greenland, implying the universal
occurrence of petit-spot and similar magmatisms (Axen et al., 2018; Buchs et al., 2013; Falloon et al.,
2022; Hirano et al., 2013, 2016, 2019; Reinhard et al., 2019; Taneja et al., 2016; Uenzelmann-Neben
et al., 2012; Yamamoto et al., 2018, 2020; Zhang et al., 2019). Although the question of whether the
LAB discontinuity is due to the differences in the physical properties of minerals (e.g., Hirth and
Kohlstedt, 1996; Kang and Karato, 2023; Karato and Jung, 1998; Katsura and Fei, 2021; Stixrude and
Lithgow-Bertelloni, 2005; Wang et al., 2006) or the presence of partial melts remains open (e.g.,
Audhkhasi and Singh, 2022; Chantel et al., 2016; Conrad et al., 2011; Debayle et al., 2020; Herath et
al., 2022; Hua et al., 2023; Kawakatsu et al., 2009; Mierdel et al., 2007; Sakamaki et al., 2013; Yoshino
et al., 2006), the occurrence of petit-spot volcanism indicates the partial melting of the asthenospheric
mantle in the region because they erupted on the seafloor without hotspot and ridge activities (Hirano
et al., 2006; Hirano and Machida, 2022; Machida et al., 2015, 2017; Yamamoto et al., 2014, 2018,

145 2020).

The petit-spot volcanic province on the abyssal plain of the western Pacific is surrounded by
Cretaceous seamounts and oceanic islands of the Western Pacific Seamount Province (Koppers et al.,
2003) and is located ~100 km southeast of the Minamitorishima (Marcus) Island (Fig. 1a). The study
area corresponds to the oldest portion of the Pacific Plate, aged at 160 Ma, and the foot of the outer-
rise bulge related to the Mariana subduction system (Hirano et al., 2019; Fig. 1b). Despite several
seamounts crosscutting, subduction-related fore-bulge in front of the Mariana Trench was detected in
satellite gravity maps and has been numerically modeled (Bellas et al., 2022; Hirano et al., 2019;
Zhang et al., 2014, 2020). Petrography, geochemistry, and geochronology of petit-spot basalts and
zircons in peperites collected from a knoll suggest that petit-spot magmas in this region ascend from
the asthenosphere along the concavely flexed plate in response to subduction into the Mariana Trench
at younger than ~3 Ma (Yamamoto et al., 2018; Hirano et al., 2019). Below the study area, a low
seismic velocity zone is observed under the lithosphere (Li et al., 2019; Fig. 1c). Notwithstanding the
low-velocity anomalies crosscutting the lower mantle (Fig. 1c), no active hotspots (i.e., heat supplies)
have been reported around the western Pacific petit-spot province , which is surrounded by Cretaceous
Wake seamount chains including Minamitorishima Island and Paleogene intraplate volcanoes
(Koppers et al., 2003; Aftabuzzaman et al., 2021; Hirano et al., 2021). Other petit-spot lava outcrops
were observed in a volcanic cluster during three research cruises using the research vessel (RV)
*Yokosuka* (YK16-01, YK18-08, and YK19-05S) with five dives using the submersible, *Shinkai* 6500
(6K#1466, 6K#1521, 6K#1522, 6K#1542, and 6K#1544; Fig. 2); and here, fresh basalts were collected.
Information related to the sampling point, depth, and thickness of palagonite rind and manganese-crust
as well as the age of the western Pacific petit-spot basalts are provided in Table 1.

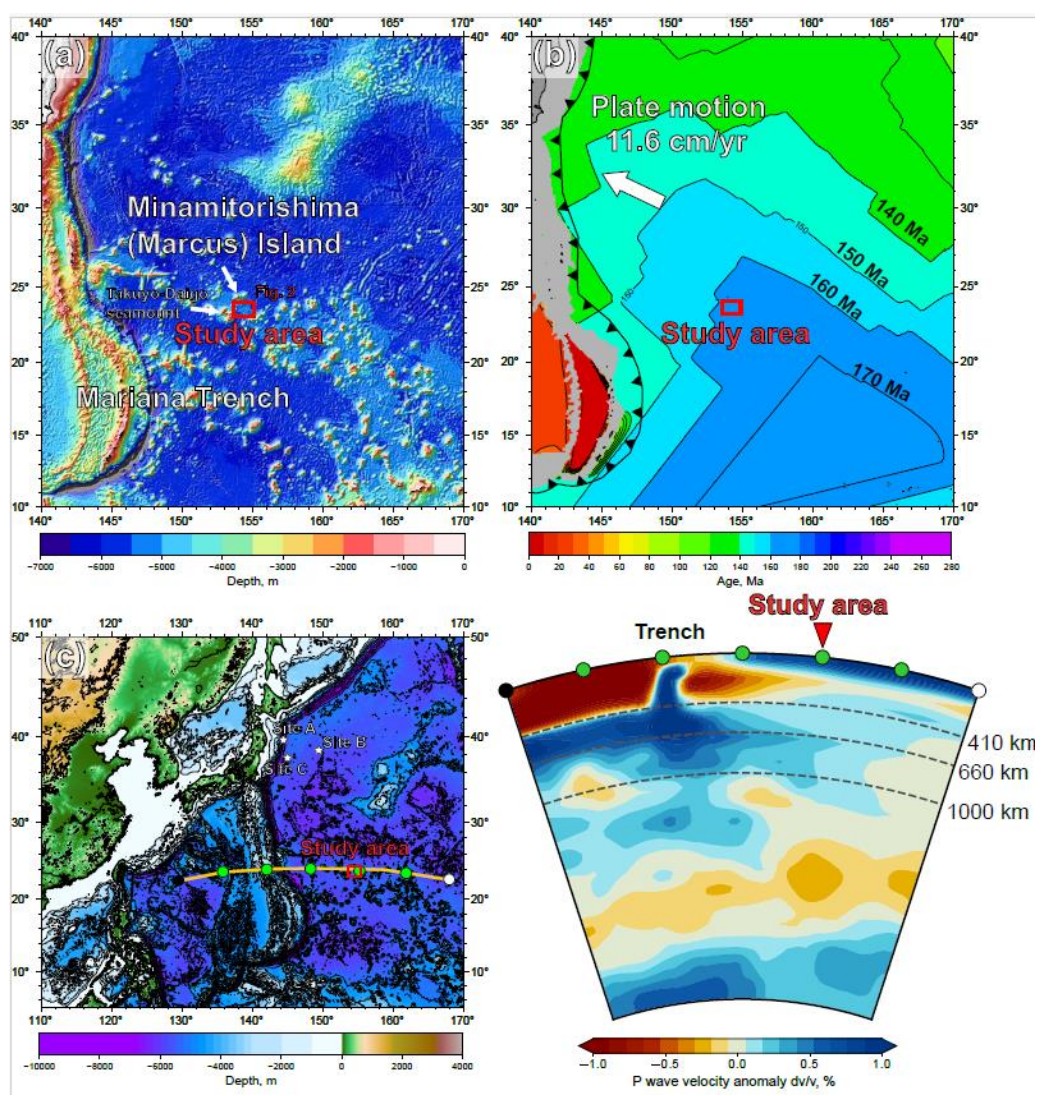


Fig. 1. Geological and geophysical information of the study area. (a) Bathymetry of the western Pacific near the
Mariana Trench. The red box shows the study area to the southeast of Minamitorishima (Marcus) Island
(Fig. 2). The bathymetric data are adopted from ETOPO1 (NOAA National Geophysical Data Center;
http://www.ngdc.noaa.gov/). (b) Seafloor age map of the same area as (a). This study area is on a 160–
170 Ma Pacific Plate, called the Jurassic Quiet Zone (JQZ) (Tivey et al. 2006). The present absolute
motion of the Pacific Plate and the seafloor age are derived from studies by Gripp and Gordon (1990)
and Müller et al. (2008), respectively. (c) The cross-section P-wave tomography beneath the thick
yellow line including the study area on the ETOPO1 bathymetry map (left). The bathymetric images
were drawn using the Generic Mapping Tool (GMT6: Wessel et al., 2019). The tomographic image
(right) was drawn using the SubMachine (Hosseini et al., 2018;
http://www.earth.ox.ac.uk/~smachine/cgi/index.php) on applying the data of Lu et al. (2019).

Table. 1
Information of the collected western Pacific petit-spot basalts

| Cruise | Dive | Sample name | Latitude (N) | Longitude (E) | Depth, m | Palagonite rind, mm [*1] | Manganese crust, mm [*1] | Ar-Ar age, Ma |
|---|---|---|---|---|---|---|---|---|
| YK16-01 | 6K#1466 | R3-001 | 23° 19.1009 | 154° 15.0950 | 5453 | 4.45 | 7.155 | |
| | | R3-04 | 23° 19.1009 | 154° 15.0950 | 5453 | 3.005 | 5.805 | |
| | | R6-001 | 23° 19.4475 | 154° 15.0367 | 5300 | 6.61 | 5.205 | 2.56±0.34 |
| | | R7-001 | 23° 19.4713 | 154° 15.0000 | 5267 | 5.54 | 4.31 | |
| | | R7-003 | 23° 19.4713 | 154° 15.0000 | 5267 | - | – | |
| YK18-08 | 6K#1521 | R04 | 23° 5.0880 | 154° 23.7360 | 5546 | 1.045 | 5.935 | |
| | | R05 | 23° 5.0880 | 154° 23.7360 | 5546 | - | 5.625 | |
| | 6K#1522 | R01 | 23° 27.6420 | 153° 58.3140 | 5300 | 6.015 | 5.78 | -0.11±0.23[*2] |
| | | R02 | 23° 27.6420 | 153° 58.3140 | 5300 | 4.505 | 2.66 | |
| | | R03 | 23° 27.6420 | 153° 58.3140 | 5300 | 5.44 | 4.04 | |
| | | R05 | 23° 27.6360 | 153° 58.3080 | 5294 | 2.92 | 4.785 | |
| | | R12 | 23° 27.4920 | 153° 58.0620 | 5189 | 6.05 | 5.56 | |
| | | R13 | 23° 27.4920 | 153° 58.0620 | 5189 | 4.545 | 5.895 | |
| | | R14 | 23° 27.3540 | 153° 57.8160 | 5303 | 2.04 | 5.475 | |
| | | R16 | 23° 27.4680 | 153° 57.1200 | 5182 | 3.825 | 3.845 | |
| | | R17 | 23° 27.4680 | 153° 57.1200 | 5182 | 5.19 | 5.67 | |
| YK19-05S | 6K#1542 | R03 | 23° 44.1926 | 154° 45.6900 | 5359 | 3.43 | 4.26 | |
| | | R05 | 23° 44.1926 | 154° 45.6900 | 5359 | 3.245 | 4.355 | |
| | | R06 | 23° 44.7064 | 154° 44.1200 | 5190 | - | - | |
| | | R09 | 23° 44.7064 | 154° 44.1200 | 5190 | - | - | |
| | 6K#1544 | R04 | 23° 43.9555 | 154° 49.4277 | 5488 | 4.39 | 4.955 | |
| | | R05 | 23° 43.9555 | 154° 49.4277 | 5488 | 2.965 | 4.97 | |
| | | R06 | 23° 43.9555 | 154° 49.4277 | 5488 | 3.425 | 5.82 | |

∗1: The samples which have no data of palagonite and/or Mn-crust thickness are due to the lack of them or crumbled.

∗2: This is a reference value due to the lack of radiogenic $^{40}Ar$ in this sample.

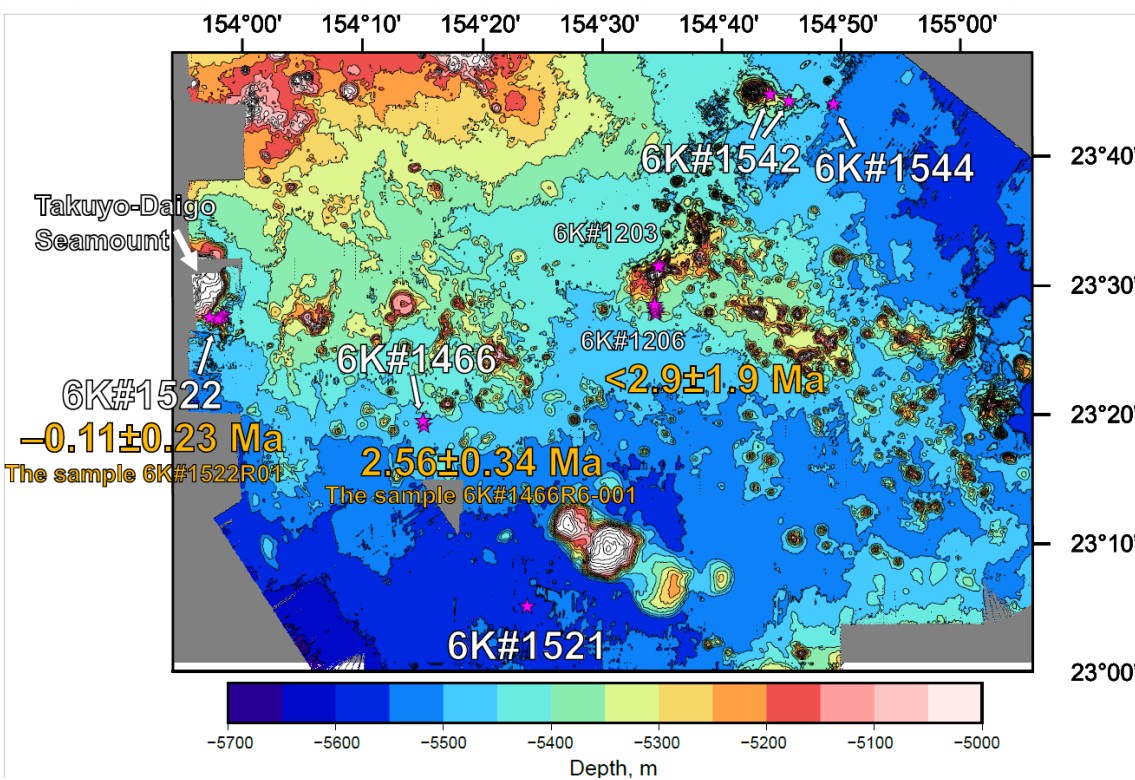

Fig. 2. Detailed bathymetry of the study area. The onboard multibeam data were surveyed during the YK10-05 and the YK18-08 cruises by the Japan Agency for Marine-Earth Science and Technology (JAMSTEC). The petit-spot knolls and outcrops were investigated during several dives as 6K#1466, 6K#1521, 6K#1522, 6K#1542, and 6K#1544. The pink-colored stars represent the sampling points. The age information was obtained in the present study and Hirano et al. (2019). The bathymetric image was drawn using the GMT (Wessel et al., 2019).

**3 Field observations, sample locations, and petrography**

Here, the eruption sites of monogenetic volcanoes or lava outcrops are approximately aligned with each dive site numbered 6K#1466, #1521, #1522, #1542, and #1544 conducted using the *Shinkai* 6500. The 6K#1466 dive was conducted at two types of monogenetic volcanoes, categorized as glassy type (R3) and crystalline and vesicular type (R6 and R7) based on the geochemical and petrographic descriptions and occurrence of basaltic samples.

**3.1 YK16-01 cruise and 6K#1466 dive**

During the YK16-01 cruise, a small conical knoll (ca. 0.04 km$^3$) was investigated by a submersible dive, 6K#1466 (Figs. 2 and 3a). The lava flows, which were observed in a hollow lava tube resulting in sediment-rolling/disturbing eruption, were located ~600 m south of the top of the knoll, featuring extremely fresh and glassy samples (6K#1466R3-001 and R3-004 basalts) (Fig. 3a). Vesicular pillow basalts were collected on the western slope of the knoll (samples 6K#1466R6-001, R7-001, and R7-003; Fig. 3a). While the strong acoustic reflection could not entirely distinguish the petit-spot lava fields in ferromanganese nodule fields, the 6K#1466 dive revealed lava outcrops using a sub-bottom profiler (SBP) and a multinarrow-beam echo sounder (MBES). Specifically, the petit-spot lava field, as an acoustically opaque layer, exhibited a vigorous backscattering intensity in the MBES, along with the distributions of the basement and sediment layers in the SBP.

The 6K#1466R3-001 and R3-004 samples were extremely fresh glassy basalts. The samples exhibited similar petrographic features (Fig. 3a). These samples were enveloped by a 3.0–4.5-mm-thick palagonite layer (hydrated quenched glass), with their outermost parts being surrounded by a 5.8–7.2-mm-thick ferromanganese crust (Fig. 3a). They were less vesicular (<3 vol.%) and were dominantly composed of basaltic glass, euhedral–subhedral olivine microphenocrysts (~100–500 μm in size), ferrotitanium oxide (<50 μm in size), and minor plagioclase (~500 μm in size) (Fig. 3a). No secondary phases such as clay minerals were observed.

The 6K#1466R6-001, R7-001, and R7-003 basalts, which were covered with a 4.3–5.2-mm-thick ferromanganese crust over 5.5–6.6-mm-thick palagonite rinds, exhibited high vesicularity (20–40 vol.%) (Fig. 3a). Mikuni et al. (2022) reported certain pyroxene-dominated xenocrysts and peridotite xenoliths. The basaltic groundmass was characterized by needle-shaped clinopyroxene (50–400 μm in size), subhedral olivine partly with aureoles of iddingsite (up to 100 μm in size), ferrotitanium oxide, minor spinel (up to 10 μm in size), glass, and crystallite, notably without remarkable phenocrysts (Fig. 3a). The photomicrograph of R6-001 is shown in Fig. 3a.

**3.2 YK18-08 cruise and 6K#1521 and 6K#1522 dives**

226

Two submersible dives (6K#1521 and #1522) were conducted during the YK18-08 cruise to investigate petit-spot volcanoes. During the 6K#1521 dive, a small lava outcrop was identified in the abyssal plain by tracing a strong acoustic reflection, which was expected to originate from intrusive rock bodies, in the sedimentary layer detected by deep-sea SBP equipped on the *Shinkai* 6500. The strong reflective surface gradually became shallow during the navigation, revealing the small lava outcrop (Figs. 2 and 3b). Fresh and massive (nonvesicular) basalts were collected from this outcrop (samples 6K#1521R04 and R05; Fig. 3b). The samples obtained from the 6K#1522 dive at a seamount exhibited highly irregular shapes, and massive lava flows, pillows, and lava breccia were observed (Fig. 3c). All the samples were fresh vesicular basalts (6K#1522R01, R02, R05, R12, R13, R16, and R17; Fig. 3c).

The fresh, massive, and nonvesicular basalts were collected during the 6K#1521 dive (R04 and R05) comprised euhedral olivine microphenocrysts (150–400 μm in size), two types of ferrotitanium oxide (50–150 μm in size), and crystallite (Fig. 2b). Secondary phases were not observed. They were covered with a 5.6–5.9-mm-thick ferromanganese crust and a ~1.0-mm-thick palagonite rind (Fig. 3b), however, R05 did not have palagonite rinds. The photomicrograph of R04 is shown in Fig. 3b.

The seven fresh basalts collected during the 6K#1522 dive (6K#1522R01, R02, R05, R12, R13, R16, and R17), exhibited high vesicularity (20–40 vol.%) with 2.9–6.0-mm-thick palagonite rinds covered with 2.7–5.9-mm-thick ferromanganese crusts (Fig. 3c). Euhedral–subhedral olivine microphenocrysts (glomeroporphyritic, 30–200 μm in size), radial–needle-shaped clinopyroxene, iddingsite (<200 μm in size), spinel, and glass with minor xenocrystic olivines were observed (Fig. 3c). The photomicrograph of R01 is shown in Fig. 3c.

**3.3 YK19-05S cruise and 6K#1542 and 6K#1544 dives**

A petit-spot knoll and associated lava flows were investigated by the 6K#1542 and #1544 dives during the YK19-05S cruise (Fig. 2). During the 6K#1542 dive, geological survey and rock sampling were conducted from two points on the eastern slope of the knoll (Figs. 2 and 3d). The 6K#1542R03 and R05 basalts were collected from the lava-breccia field covered with a thin ferromanganese crust (Fig. 3d). Additionally, samples R06 and R09 were obtained from the lobate-surface lava between tubular lavas closer to the summit than R03 and R05 (Fig. 3d).

High-resolution (one-meter scale) bathymetric mapping was successfully conducted during the 6K#1544 dive, which can contribute to future oceanographic investigations using a human-occupied vehicle (Kaneko et al., 2022). Several mounds, 10–20 m in height and a few hundred meters in diameter, were recognized during this acoustic survey (Fig. 3d). We observed these mounds and collected samples from outcrops during the second half of the dive. Furthermore, pillow lavas, tumuli,

and lava breccias were observed, and basaltic samples (6K#1544R04, R05, and R06) were collected
(Fig. 3d).

264     Four vesicular basalts (10–30 vol.% vesicularity; 6K#1542R03, R05, R06, and R09) were
covered with 4.3–4.4-mm-thick ferromanganese crusts. The outer palagonitic rinds were 3.2–3.4-mm-
thick (Fig. 3d). Euhedral–subhedral olivine microlites (up to sizes of 300 μm) and microphenocrysts
were glomeroporphyritic (Fig. 3d). The groundmass was dominated by needled dendritic
clinopyroxenes (~100 μm in size), along with olivine, spinel, glass, and xenocrystic olivine megacrysts.
The photomicrograph of R06 is shown in Fig. 3d.

270     Basaltic samples from the 6K#1544 dive (6K#1544R04, R05, and R06) were covered with
ferromanganese crust (5.0–5.8-mm thick) over palagonitic rinds (3.4–4.4-mm thick). All the samples
exhibited high vesicularity in the range of 20–35 vol.% (Fig. 3d). They comprised olivine
microphenocrysts (30–250 μm in size, euhedral–subhedral or columnar), clinopyroxene (<100 μm,
needled, columnar, radial or dendritic shape), spinel, and glass without secondary phases (Fig. 3d).

275     The photomicrograph of R04 is shown in Fig. 3d. During macroscopic observations, practically
all the basalts from the 6K#1542 and 6K#1544 dives exhibited similar vesicularity and freshness.
Their geochemical features were also similar to each other and are described in Sect. 5-1 and 5-2.

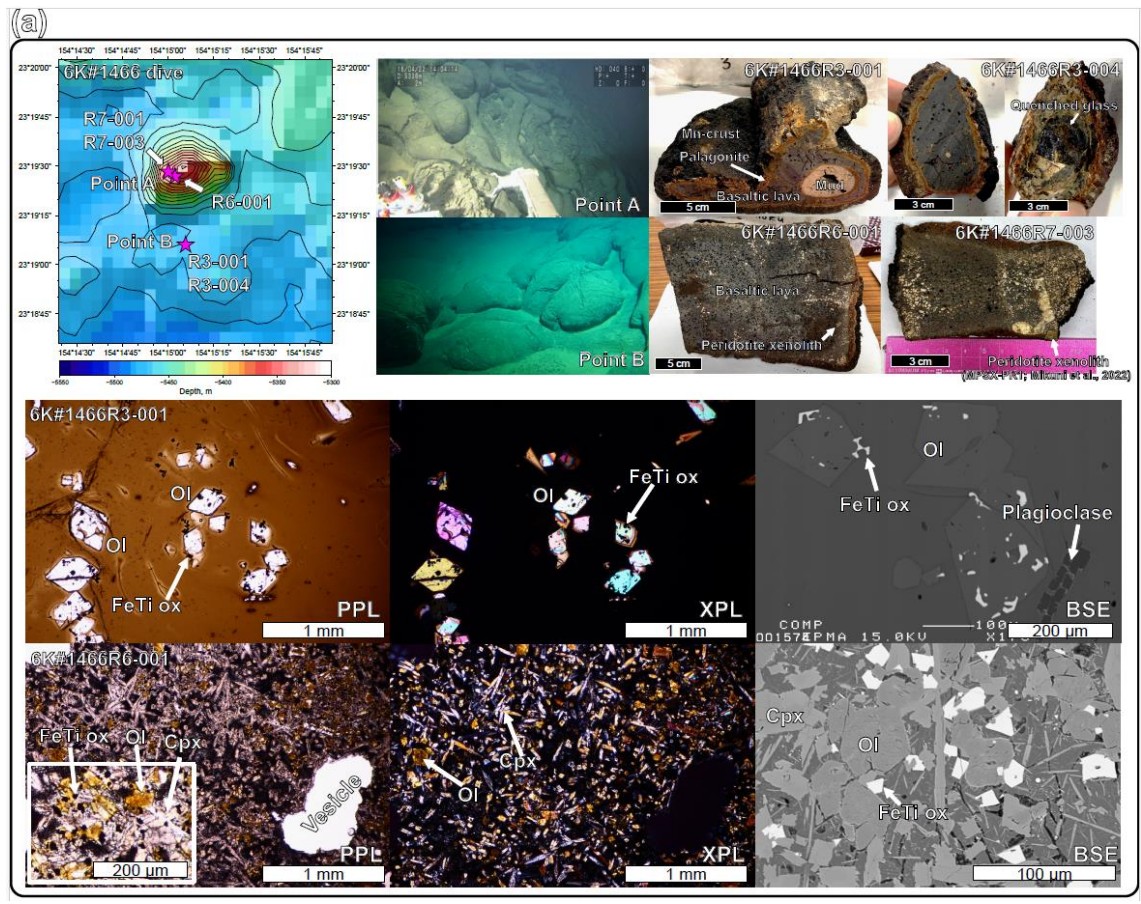



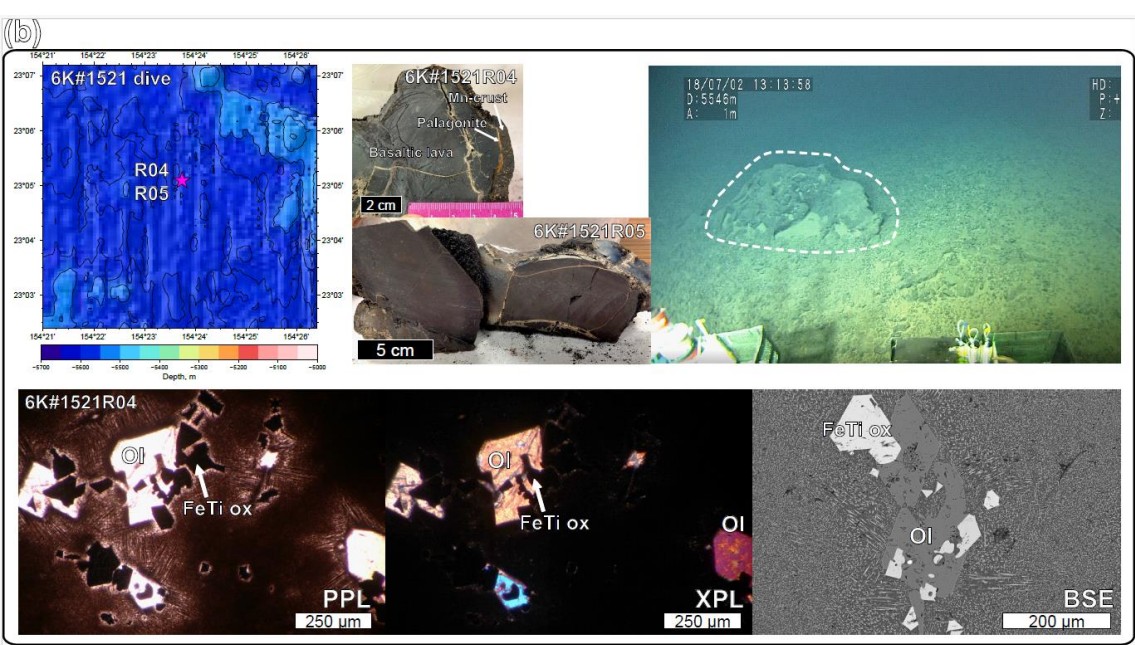


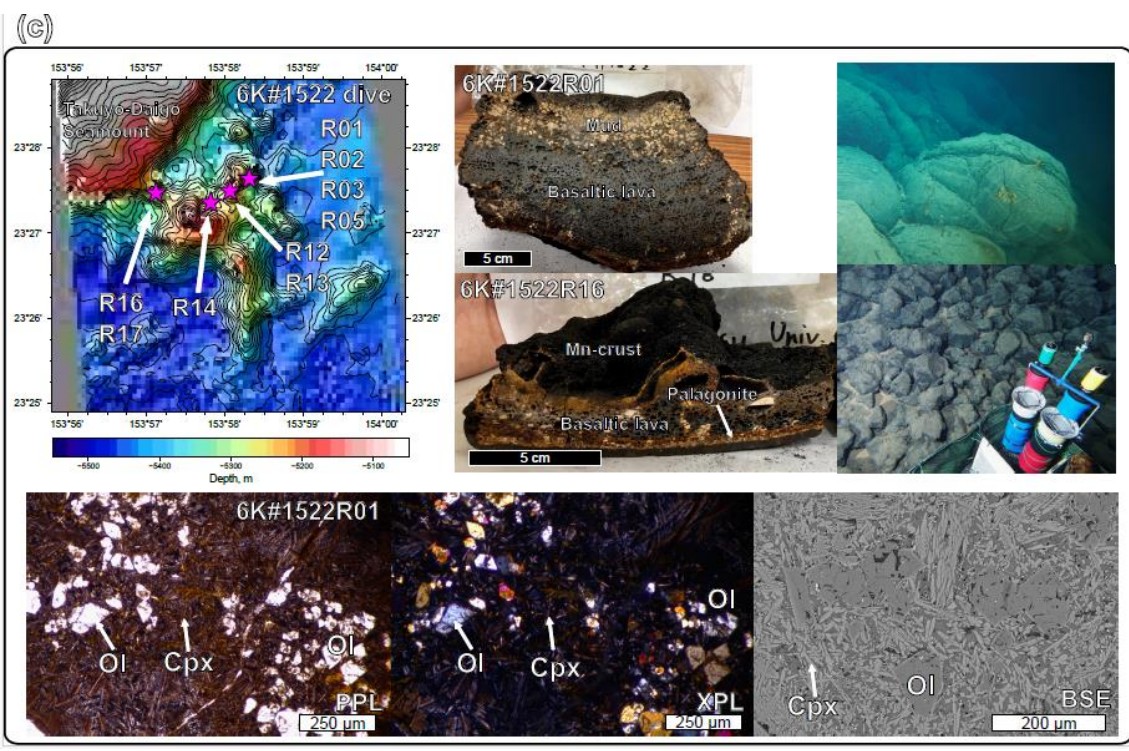

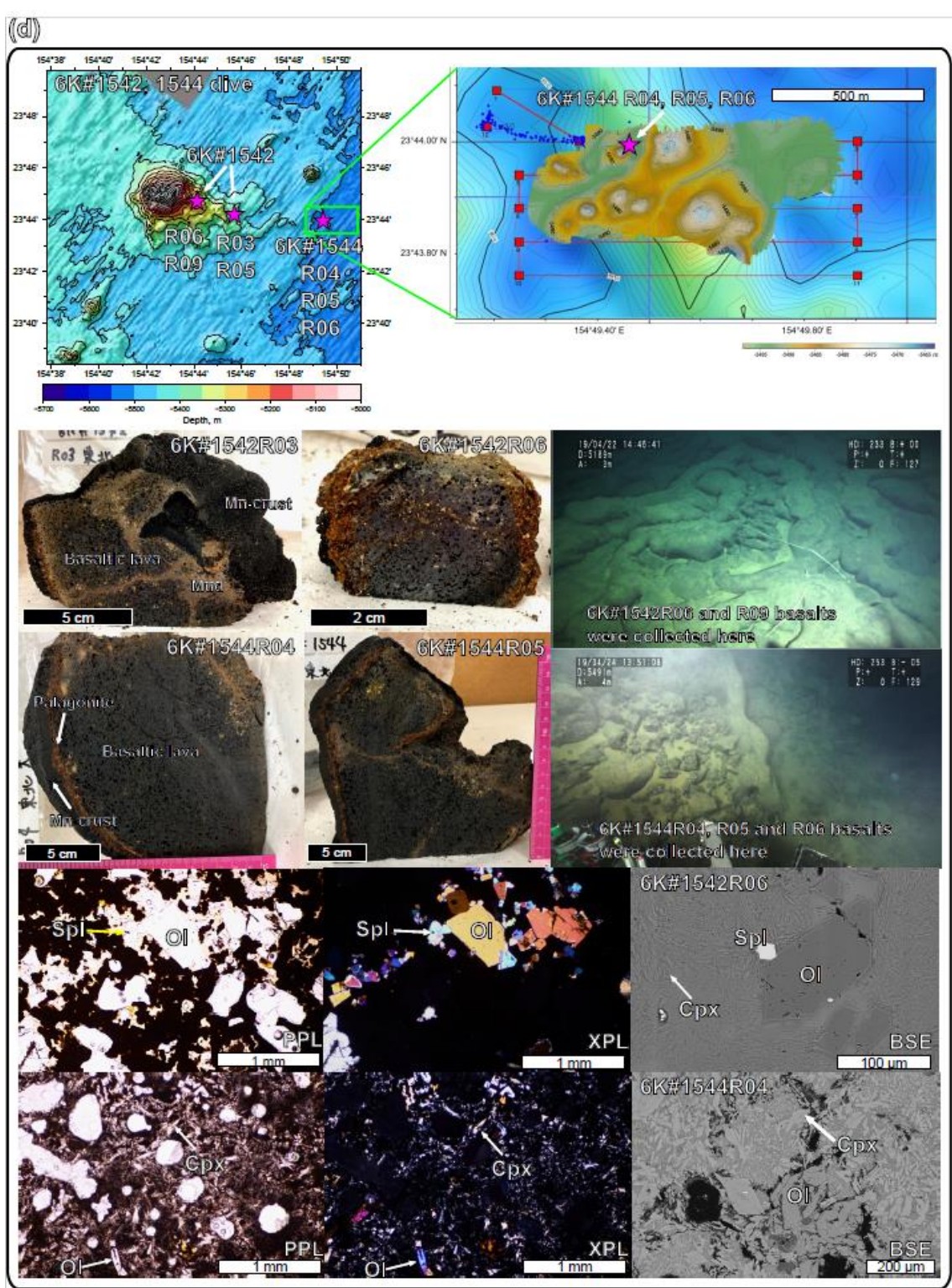

Fig. 3. Bathymetric map with photos of the outcrop, the collected samples, and their photomicrographs with detailed bathymetry of the sampling points. (a) The 6K#1466, (b) 6K#1521, (c) 6K#1522, and (d) 6K#1542 and 6K#1544 dives using the *Shinkai* 6500 by JAMSTEC. The 1-m gridded bathymetry of the 6K#1544 dive is shown in (d), obtained using an MBES equipped with the *Shinkai* 6500 over a 100-m resolution map

obtained using the surface ship, R/V *Yokosuka* (Kaneko et al., 2022). The photomicrographs of
representative samples are shown for plane-polarized light (PPL), cross-polarized light (XPL), and
backscatter electron (BSE). Ol, olivine; Cpx, clinopyroxene; Mgt, magnetite; Spl, spinel. The bathymetric
images were drawn using the GMT (Wessel et al., 2019). Photos of seafloor lava-outcrops were provided
by the cruise report of YK16-01, YK18-08, and YK19-05S cruises in the Data and Sample Research
System for Whole Cruise Information by JAMSTEC (http://www.godac.jamstec.go.jp/darwin/).


**4. Analytical methods**

**4.1 Major and trace element analysis of volcanic glass, mineral, and whole-rock**

Major element compositions of glasses and minerals were determined using an electron probe
micro analyzer (EPMA). JXA-8900R at Atmosphere and Ocean Research Institute (AORI), the
University of Tokyo was used for glass analysis and JXA-iHP200F at GSJ, AIST was used for mineral
analysis. The analyses were performed using an accelerating voltage of 15 kV, a beam current of 12
nA, and a beam diameter of 10 μm for glass and 2 μm for mineral. A peak counting time of 20 s and
a background counting time of 10 s were used, except for Ni, for which a peak counting time of 30 s
and a background counting time of 15 s. For Na analysis of glass, the peak counting time was 5 s and
the background counting time was 2 s. Natural and synthetic minerals were used as standards, and data
were corrected using a ZAF online correction program (Akizawa et al., 2021). Major element
composition of glass was determined by the mean value of 10 analytical points.
Trace element compositions of minerals were determined using a laser ablation-inductively
coupled plasma-mass spectrometry (LA-ICP-MS; New Wave Research UP-213 and Agilent 7500s)
at Kanazawa University. The Nd: YAG deep UV (ultraviolet) laser's wavelength is 213 nm. The
analyses were conducted with 100 μm spot size. A repetition frequency of 6 Hz and a laser energy
density of 8 J cm$^{-2}$ were used. NIST612 glass (distributed by National Institute of Standards and
Technology) was employed for calibration, using the preferred values of Pearce et al. (1997). Data
reduction was undertaken with $^{29}$Si as the initial standard, and SiO$_2$ concentrations were obtained by
an electron microprobe analysis (Longerich et al., 1996). BCR-2G (distributed by the United States
Geological Survey) was used as a secondary standard to assess the precision of each analytical
session (Jochum and Nohl, 2008).
Whole-rock major and trace element compositions of rock samples were analyzed by Activation
Laboratories Ltd., Canada, using Code 4Lithoresearch Lithogeochemistry and ultratrace5 Exploration
Geochemistry Package. The former package uses lithium metaborate/tetraborate fusion with
inductively coupled plasma optical emission spectrometry (FUS-ICP-OES) and inductively coupled

plasma mass spectroscopy (FUS-ICP-MS) for the major and trace element analyses, respectively. The latter package uses inductively coupled plasma optical emission spectrometry (ICP-OES) and inductively coupled plasma mass spectroscopy (ICP-MS) for the major and trace element analyses, respectively.

**4.2 Sr, Nd, and Pb isotope analysis**

**4.2.1 Acid leaching**

Acid leaching was conducted for the selected basaltic samples on the basis of the procedure of Weis and Frey (1991, 1996) as follows: [1] About 0.3–0.4 or 0.6 g of rock powder is weighed into an acid-washed 15 mL Teflon vial (Savilex®). [2] 10 or 12 mL of 6N (N: normality) HCl were added, and then heated at 80°C for 20–30 min. [3] After heating, the suspension is ultra-sonicated in 60°C water for 20 min. [4] The supernatant is decanted. Steps [2] to [4] were repeated more than 4 times (up to 6 times) until the supernatant become clear or pale yellow to colorless. [5] TAMAPURE-AA Ultrapure water (Tama Chemicals; Co., Ltd.), which includes a lower Pb blank than milli-Q $H_2O$, were added instead of 6N HCl, and the suspension is ultra-sonicated for 20 min. This step is conducted twice. [6] The leached rock powder is dried on a hot plate at 120°C. [7] After cooling, the powder is weighed.

**4.2.2 Extraction of Pb, Sr, and Nd**

The extraction of Pb, Sr, and Nd was performed following the procedures of Tanimizu and Ishikawa (2006) and Machida et al. (2009). First, from ~50 to ~100 mg of rock powder was weighted in a 7 mL Teflon vial (designated as "vial A"), and digested using mixed acid composed of HF and HBr. The separation was conducted by cation exchange resin (AG-1X8; Bio-Rad Laboratories Inc.) on the basis of procedures described in Tanimizu and ishikawa (2006). All fractions from the first and second supernatant loading (0.5 M HBr) to the elution of other elements (mixed acid composed of 0.25 M HBr and 0.5 M $HNO_3$) were collected in another 7 mL Teflon vial (designated as "vial B") for Sr and Nd separation. Finally, Pb was extracted by 1 mL of 1M $HNO_3$ in another 7 mL Teflon vial (designated as "vial C"). The procedural blanks for Pb totaled less than 23 pg.

The Sr and Nd-bearing solution in the vial B was transferred into the vial A containing residues of digested samples. 2 mL of $HClO_4$ and 2 mL $HNO_3$ was further added to the vial A, and the residue was dissolved at 110 °C. Both Sr and Nd were separated by column with a cation exchange resin (AG50W-8X; Bio-Rad Laboratories Inc.) and a Ln resin (Eichrom Tech- nologies Inc.) on the basis of procedures described in Machida et al. (2009). The separated Sr and Nd were further purified by column separation with a cation exchange resin. The total procedural blanks for Sr and Nd were less

than 100 pg.

**4.2.3 Analytical procedure**

Pb isotopic ratios were obtained using the multi-collector ICP-MS (MC-ICP-MS; Neptune plus,
Thermo Fisher Scientific), with nine Faraday collectors, at Chiba Institute of Technology (CIT), Japan.
The NIST SRM-981 Pb standard was also analyzed and yielded the average values of $^{206}Pb/^{204}Pb =$
$16.9303 \pm 0.0005$, $^{207}Pb/^{204}Pb = 15.4828 \pm 0.0006$, and $^{208}Pb/^{204}Pb = 36.6710 \pm 0.0016$. These
correspond to previous values determined using MC-ICP-MS with Tl normalization, but they were
slightly lower than values determined by TIMS in Tanimizu and Ishikawa (2006) from the $^{207}Pb–^{204}Pb$
double-spike. Reproducibility was monitored by an analyses of the JB-2 GSJ standard, and the
obtained values were $^{206}Pb/^{204}Pb = 18.3326 \pm 0.0005$, $^{207}Pb/^{204}Pb = 15.5453 \pm 0.0006$, and $^{208}Pb/^{204}Pb$
$= 38.2240 \pm 0.0017$.
Sr and Nd isotopic analyses for powdered rocks and glasses were conducted using the thermal
ionization mass spectrometry (TIMS; Triton XT, Thermo Fisher Scientific) with nine Faraday
collectors, at CIT. 1.5 µL of 2.5M HCl and 0.5M $HNO_3$ was used for loading of separated Sr and Nd
of sample on the single and double Re-filament, respectively. The measured isotopic ratios were
corrected for instrumental fractionation by adopting the $^{86}Sr/^{85}Sr$ value to be 0.1194 and that of
$^{146}Nd/^{144}Nd$ to be 0.7219. The average value for the NIST SRM-987 Sr standard was 0.710239
$\pm 0.000005$ (2σ, n =2), and that for the GSJ JNdi-1 Nd standard was 0.512103 $\pm 0.000005$ (2σ, n =2).
They agree well with values from the literature for the NIST SRM-987 ($^{87}Sr/^{86}Sr = 0.710252–$
0.710256; Weis et al., 2006) and JNdi-1 ($^{143}Nd/^{144}Nd = 0.512101$; Wakaki et al., 2007). Consequently,
we did not correct the values of the unknowns for offsets between the measurements and the values
for the Sr and Nd standards.

**4.3 $^{40}Ar/^{39}Ar$ dating**

Samples for $^{40}Ar/^{39}Ar$ dating were prepared by separating crystalline groundmass after crushing
them to sizes between 100 and 500 µm. The separated groundmass samples were leached by $HNO_3$ (1
mol/L) for one hour to remove clays and altered materials. All samples were wrapped in aluminum
foil along with JG-1 biotite (Iwata, 1998), $K_2SO_4$, and $CaF_2$ flux monitors. Any amorphous (e.g.,
quenched glass) was removed because $^{39}Ar$ may move from one phase to another in a process known
as "recoil." This can create a disturbed age spectrum when $^{39}Ar$ is produced from $^{39}K$ in amorphous
material through interaction with fast neutrons during irradiation of the sample. Samples were
irradiated for 6.6 days in the Kyoto University Research Reactor (KUR), Kyoto University. Argon
extraction and isotopic analyses were undertaken at the Graduate School of Arts and Sciences, the
University of Tokyo. The sample gases were extracted by incremental heating of 10 or 11 steps
between 600°C and 1500°C. The analytical methods used are the same as those used by Ebisawa et al.
(2004) and Kobayashi et al. (2021).

**5 Results**

To describe the geochemical and chronological results, each sample group was denoted by its
dive number, e.g., the sample group obtained from the 6K#1521 dive was labeled "1521 samples or
basalts". The basalts from the 6K#1466 dive were divided into two groups for R3 (collected from the
seafloor south of the knoll) and R6–R7 (sampled on the knoll) based on their geographical,
petrological, and compositional differences. The mineral compositions of each petit-spot basalt are
shown in Fig. S1 and Table S1, S2 and S3.

**5.1 Major and trace element compositions**

The major and trace element compositions for the whole rock and glass of the petit-spot basalts
are listed in Table 2 and 3, respectively. The basalt compositions for a petit-spot knoll were reported
by Hirano et al. (2019) (expressed as "1203, 1206" in each figure). The data are discussed along with
the reported NW Pacific petit-spots (Hirano and Machida, 2022). Using a total alkali vs. silica (TAS)
diagram, virtually all the samples were classified as alkalic rocks, but the 1542 and 1544 basalts were
plotted near the boundary between alkalic and non-alkalic (Fig. 4a). Two petit-spot basalts (1466R7-
001 and R7-003) from the petit-spot knoll were notably silica-undersaturated (i.e., $SiO_2$ = 39.3–39.4
wt%) and classified as foidite (Mikuni et al., 2022). All the western Pacific petit-spot basalts, except
for the 1466R7 basalts, were sodic ($K_2O/Na_2O$ = 0.24–0.58) and were notably discriminated to the
potassic NW Pacific petit-spots (Fig. 4b).
Selected major element oxides and trace element ratios vs. MgO plots for the petit-spot basalts
are shown in Figs. 5 and 6, respectively. The MgO concentrations of the 1466R3 and 1521 samples
each exhibiting similar petrographic features (i.e., nonvesicular, and glassy) were characterized by
values (4.0–4.4 wt%) lower than those of other vesicular samples (6.6–9.3 wt%). The $K_2O$, $Na_2O$,
$Al_2O_3$, and $SiO_2$ contents negatively correlated with MgO (Figs. 5a–d). The CaO, $FeO_T$, and
$CaO/Al_2O_3$ abundances exhibited positive correlations with MgO (Figs. 5e–g). The $TiO_2$
concentrations exhibited no correlations with MgO (Fig. 5h), as well as the selected trace element
ratios (Figs. 6a–g) except for the Sm/Hf ratio with positive correlations (Fig. 6h). The Sm/Hf ratio also
negatively correlated with $SiO_2$ (Fig. S2). The study samples exhibited whole-rock loss on ignition
(LOI) in the range of 0.67–1.72 wt%, excluding two relatively altered samples, 1466R7-001 (LOI =
2.68 wt%) and R7-003 basalts (LOI = 6.29 wt%).

The PM-normalized (Sun and McDonough, 1989) trace element patterns for the petit-spot basalts, including those reported by a previous study (Hirano et al., 2019), were shown for each dive compared to the representative ocean island basalt (OIB) in Figs. 7a–f. The petit-spot basalts generally showed high light rare earth element (LREE)/heavy REE (HREE) ratios. Negative Zr, Hf, Ti, and Y anomalies were commonly observed in these western Pacific petit-spots as well as those of the NW Pacific petit-spots (Fig. 7g). The 1466 basalts collected on the seafloor south of the knoll (1466R3-001 and 1466R3-004 basalts) were compositionally different from those obtained on the knoll (1466R7-001 and 1466R7-003 samples). The basalts from the 6K#1542 and #1544 dives, collected from nearby locations, had the same compositions in major and trace element ratios in both whole rock and glass, respectively (Figs. 4, 5, 6, 7e, and f). These samples in the Ba/Nb and Sm/Hf diagrams were plotted in the range of "Group 3" in the discrimination of the NW Pacific petit-spot basalts (Machida et al., 2015), indicating their negative Zr and Hf anomalies without notable U, Th, Nb, and Ta anomalies in the PM-normalized trace element patterns (Fig. 7h). The Sm/Hf ratio of the differentiated 1466R3 samples was lower than that of other samples. A positive correlation between fluid mobile and immobile elements, Ba vs. Nb (Fig. 8a) and U vs. Th (Fig. 8b), respectively, was observed, excluding the Ba of the 1466R7 samples (Fig. 8a).

Table. 2
Major and trace element compositions of western Pacific petit-spot basalts.

| Cruise | YK16-01 | | YK16-01 | | YK16-01 | YK16-01 | YK18-08 | | YK18-08 | | YK18-08 | | YK18-08 | YK18-08 | | YK18-08 | | YK18-08 | |
|---|---|---|---|---|---|---|---|---|---|---|---|---|---|---|---|---|---|---|---|
| Sample name | 6K#1466R3-001 | | 6K#1466R3-004 | | 6K#1466R7-001 | 6K#1466R7-003 | 6K#1521R04 | | 6K#1521R05 | | 6K#1522R01 | | 6K#1522R01 | 6K#1522R02 | | 6K#1522R05 | | 6K#1522R12 | |
| Sample type | Glass | | Glass | | Whole rock | Whole rock | Glass | | Glass | | Glass | | Whole rock | Glass | | Glass | | Glass | |
| Method | EPMA | | EPMA | | * | * | EPMA | | EPMA | | EPMA | | * | EPMA | | EPMA | | EPMA | |
| | mean of n=10 | 2σ | mean of n=10 | 2σ | | | mean of n=10 | 2σ | mean of n=10 | 2σ | mean of n=10 | 2σ | | mean of n=10 | 2σ | mean of n=10 | 2σ | mean of n=10 | 2σ |
| wt% | | | | | | | | | | | | | | | | | | | |
| SiO$_2$ | 51.56 | 0.93 | 50.63 | 0.79 | 39.40 | 39.27 | 48.42 | 0.36 | 46.78 | 0.97 | 45.92 | 1.40 | 45.28 | 45.90 | 0.79 | 45.38 | 1.56 | 46.02 | 0.69 |
| TiO$_2$ | 2.31 | 0.20 | 2.19 | 0.22 | 3.82 | 3.68 | 3.65 | 0.04 | 3.32 | 0.25 | 2.37 | 0.17 | 2.43 | 2.51 | 0.20 | 2.33 | 0.13 | 2.45 | 0.21 |
| Al$_2$O$_3$ | 14.99 | 0.57 | 15.10 | 0.37 | 11.41 | 11.46 | 15.12 | 0.31 | 14.38 | 0.45 | 12.74 | 0.23 | 12.48 | 12.82 | 0.25 | 11.99 | 0.53 | 12.91 | 0.14 |
| Cr$_2$O$_3$ | - | | - | | 0.03 | 0.03 | - | | - | | 0.01 | 0.05 | 0.03 | 0.02 | 0.05 | 0.01 | 0.05 | 0.02 | 0.04 |
| FeO$^T$ | 9.68 | 0.30 | 9.17 | 0.62 | 15.12 | 14.90 | 10.65 | 0.29 | 9.77 | 0.79 | 11.72 | 0.16 | 12.32 | 11.64 | 0.42 | 10.77 | 1.02 | 11.62 | 0.24 |
| MnO | 0.14 | 0.04 | 0.14 | 0.05 | 0.21 | 0.20 | 0.16 | 0.04 | 0.14 | 0.03 | 0.18 | 0.04 | 0.18 | 0.16 | 0.04 | 0.15 | 0.05 | 0.17 | 0.05 |
| MgO | 4.04 | 0.11 | 3.99 | 0.11 | 9.34 | 7.66 | 4.43 | 0.08 | 4.36 | 0.10 | 7.36 | 0.17 | 7.26 | 7.33 | 0.10 | 7.12 | 0.23 | 7.14 | 0.16 |
| CaO | 7.71 | 0.11 | 7.41 | 0.25 | 11.19 | 10.02 | 8.34 | 0.68 | 7.80 | 0.29 | 10.72 | 0.14 | 11.18 | 10.81 | 0.22 | 10.33 | 0.68 | 10.79 | 0.14 |
| Na$_2$O | 4.61 | 0.24 | 4.38 | 0.50 | 2.15 | 2.29 | 3.84 | 0.31 | 4.05 | 0.55 | 4.16 | 0.21 | 3.53 | 4.16 | 0.29 | 4.16 | 0.24 | 4.01 | 0.46 |
| K$_2$O | 2.31 | 0.08 | 2.24 | 0.12 | 1.65 | 2.08 | 2.25 | 0.27 | 2.13 | 0.12 | 1.38 | 0.06 | 1.42 | 1.40 | 0.13 | 1.31 | 0.10 | 1.38 | 0.04 |
| NiO | 0.01 | 0.03 | 0.01 | 0.03 | 0.03 | 0.02 | - | 0.04 | - | 0.05 | 0.02 | 0.03 | 0.02 | 0.01 | 0.04 | 0.02 | 0.04 | 0.02 | 0.04 |
| P$_2$O$_5$ | 0.93 | 0.03 | 0.91 | 0.06 | 1.08 | 1.12 | 1.53 | 0.11 | 1.51 | 0.03 | 0.80 | 0.06 | 0.83 | 0.80 | 0.08 | 0.82 | 0.06 | 0.77 | 0.04 |
| Total | 98.28 | | 96.16 | | 98.10 | 99.02 | 98.38 | | 94.24 | | 97.35 | | 98.67 | 97.56 | | 94.40 | | 97.31 | |
| Mg# | 42.64 | | 43.68 | | 52.42 | 47.82 | 42.57 | | 44.33 | | 52.83 | | 51.24 | 52.89 | | 54.11 | | 52.28 | |
| LOI | | | | | 2.68 | 6.29 | | | | | | | 1.72 | | | | | | |

FeO$^T$ as total values.
Mg# = 100 x Mg / [Mg+Fe$^{2+}$]$_{molar}$.
" - ": not detected
*: Analyzed by ActLab

Table. 2 continued

| | YK18-08 | | YK18-08 | | YK18-08 | | YK19-05S | | YK19-05S | YK19-05S | | YK19-05S | | YK19-05S | | YK19-05S | | YK19-05S | YK19-05S | | YK19-05S | |
|---|---|---|---|---|---|---|---|---|---|---|---|---|---|---|---|---|---|---|---|---|---|---|
| | 6K#1522R13 | | 6K#1522R16 | | 6K#1522R17 | | 6K#1542R03 | | 6K#1542R03 | 6K#1542R05 | | 6K#1542R06 | | 6K#1542R09 | | 6K#1544R04 | | 6K#1544R04 | 6K#1544R05 | | 6K#1544R06 | |
| | Glass | | Glass | | Glass | | Glass | | Whole rock | Glass | | Glass | | Glass | | Glass | | Whole rock | Glass | | Glass | |
| | EPMA | | EPMA | | EPMA | | EPMA | | * | EPMA | | EPMA | | EPMA | | EPMA | | * | EPMA | | EPMA | |
| | mean of n=10 | 2σ | mean of n=10 | 2σ | mean of n=10 | 2σ | mean of n=10 | 2σ | | mean of n=10 | 2σ | mean of n=10 | 2σ | mean of n=10 | 2σ | mean of n=10 | 2σ | | mean of n=10 | 2σ | mean of n=10 | 2σ |
| | | | | | | | | | | | | | | | | | | | | | | |
| | 47.09 | 0.68 | 45.22 | 0.73 | 45.06 | 0.98 | 48.66 | 1.14 | 49.35 | 48.77 | 1.51 | 49.66 | 1.11 | 50.09 | 0.93 | 50.54 | 0.43 | 49.08 | 50.53 | 0.61 | 49.59 | 1.18 |
| | 2.50 | 0.20 | 2.58 | 0.20 | 2.67 | 0.27 | 2.11 | 0.19 | 2.16 | 2.13 | 0.18 | 2.25 | 0.22 | 2.24 | 0.20 | 2.04 | 0.23 | 2.13 | 2.08 | 0.25 | 2.07 | 0.24 |
| | 13.08 | 0.33 | 12.55 | 0.17 | 12.55 | 0.14 | 13.49 | 0.18 | 12.52 | 13.38 | 0.19 | 12.55 | 0.43 | 12.78 | 0.33 | 13.18 | 0.12 | 13.25 | 12.94 | 0.34 | 12.94 | 0.36 |
| | 0.02 | 0.05 | 0.01 | 0.04 | 0.02 | 0.08 | 0.04 | 0.05 | 0.05 | 0.03 | 0.07 | 0.02 | 0.04 | 0.04 | 0.04 | 0.03 | 0.05 | 0.05 | 0.03 | 0.05 | 0.03 | 0.04 |
| | 11.74 | 0.49 | 11.94 | 0.40 | 11.89 | 0.26 | 10.60 | 0.30 | 11.40 | 10.47 | 0.36 | 10.22 | 0.51 | 10.44 | 0.34 | 10.46 | 0.34 | 11.13 | 10.77 | 0.37 | 10.53 | 0.49 |
| | 0.17 | 0.05 | 0.18 | 0.05 | 0.18 | 0.05 | 0.15 | 0.04 | 0.17 | 0.14 | 0.04 | 0.15 | 0.04 | 0.16 | 0.04 | 0.16 | 0.02 | 0.16 | 0.16 | 0.05 | 0.15 | 0.05 |
| | 6.63 | 0.64 | 7.24 | 0.17 | 7.24 | 0.17 | 7.29 | 0.17 | 8.18 | 7.29 | 0.22 | 7.03 | 0.13 | 7.11 | 0.12 | 7.00 | 0.16 | 7.50 | 7.10 | 0.15 | 7.05 | 0.15 |
| | 11.01 | 0.25 | 11.17 | 0.24 | 11.19 | 0.25 | 10.03 | 0.14 | 10.74 | 10.00 | 0.10 | 9.90 | 0.32 | 10.03 | 0.24 | 10.63 | 0.26 | 10.67 | 10.36 | 0.17 | 10.33 | 0.22 |
| | 4.16 | 0.36 | 4.30 | 0.33 | 4.28 | 0.39 | 3.30 | 0.28 | 2.59 | 3.36 | 0.24 | 3.39 | 0.19 | 3.26 | 0.46 | 3.54 | 0.25 | 2.90 | 3.52 | 0.26 | 3.42 | 0.28 |
| | 1.42 | 0.17 | 1.52 | 0.08 | 1.51 | 0.06 | 0.80 | 0.05 | 0.77 | 0.80 | 0.06 | 0.89 | 0.04 | 0.91 | 0.06 | 0.85 | 0.08 | 0.85 | 0.85 | 0.06 | 0.83 | 0.04 |
| | 0.01 | 0.04 | 0.01 | 0.04 | 0.01 | 0.04 | 0.01 | 0.05 | 0.02 | 0.02 | 0.05 | 0.02 | 0.05 | 0.03 | 0.05 | 0.02 | 0.03 | 0.02 | 0.01 | 0.04 | 0.02 | 0.04 |
| | 0.83 | 0.05 | 0.95 | 0.07 | 0.95 | 0.03 | 0.48 | 0.04 | 0.50 | 0.50 | 0.04 | 0.51 | 0.04 | 0.52 | 0.06 | 0.54 | 0.03 | 0.52 | 0.57 | 0.05 | 0.55 | 0.04 |
| | 98.66 | | 97.67 | | 97.54 | | 96.96 | | 99.12 | 96.91 | | 96.62 | | 97.60 | | 98.98 | | 99.09 | 98.91 | | 97.50 | |
| | 50.18 | | 51.93 | | 52.04 | | 55.07 | | 56.13 | 55.38 | | 55.07 | | 54.83 | | 54.39 | | 54.57 | 54.04 | | 54.41 | |
| | | | | | | | | | 0.67 | | | | | | | | | 0.83 | | | | |

Table 3

| µg/g | YK16-01 6K#1466R3-001 Glass LA-ICPMS | YK16-01 6K#1466R3-004 Glass LA-ICPMS | YK16-01 6K#1466R7-001 Whole rock * | YK16-01 6K#1466R7-003 Whole rock * | YK18-08 6K#1521R04 Glass LA-ICPMS | YK18-08 6K#1521R05 Glass LA-ICPMS | YK18-08 6K#1522R01 Glass LA-ICPMS | YK18-08 6K#1522R01 Whole rock * | YK18-08 6K#1522R02 Glass LA-ICPMS | YK18-08 6K#1522R05 Glass LA-ICPMS | YK18-08 6K#1522R12 Glass LA-ICPMS |
|---|---|---|---|---|---|---|---|---|---|---|---|
| Li | 7.60 | 7.32 | | | 7.39 | 7.00 | 8.10 | | 7.69 | 7.83 | 7.71 |
| B | 2.92 | 3.17 | | | 3.05 | 3.48 | 2.38 | | 2.34 | 2.78 | 2.69 |
| Sc | 14.9 | 15.2 | 25.0 | 25.0 | 15.7 | 15.4 | 20.1 | 20.6 | 20.6 | 21.2 | 21.1 |
| V | 159 | 160 | 353 | 324 | 167 | 157 | 204 | 234 | 208 | 207 | 207 |
| Cr | 36.8 | 37.1 | 200 | 190 | 0.52 | 0.48 | 215 | 190 | 218 | 213 | 222 |
| Co | 29.7 | 29.9 | 61.0 | 57.0 | 32.8 | 31.2 | 46.2 | 49.0 | 46.8 | 46.1 | 47.3 |
| Rb | 47.5 | 47.6 | 26.0 | 32.0 | 34.1 | 33.4 | 25.8 | 28.0 | 26.9 | 26.8 | 26.6 |
| Sr | 976 | 991 | 577 | 307 | 1385 | 1361 | 848 | 827 | 924 | 943 | 901 |
| Y | 21.8 | 22.2 | 37.0 | 58.0 | 33.1 | 32.2 | 24.4 | 25.0 | 26.0 | 27.6 | 26.7 |
| Zr | 254 | 260 | 259 | 248 | 293 | 286 | 157 | 163 | 168 | 177 | 171 |
| Nb | 56.4 | 57.5 | 65.0 | 64.0 | 58.7 | 57.6 | 49.5 | 52.0 | 55.3 | 55.7 | 54.6 |
| Cs | 0.58 | 0.58 | - | - | 0.35 | 0.34 | 0.32 | - | 0.35 | 0.37 | 0.34 |
| Ba | 613 | 623 | 453 | 317 | 577 | 565 | 447 | 479 | 512 | 528 | 500 |
| La | 44.1 | 45.4 | 65.2 | 90.8 | 44.2 | 42.8 | 42.8 | 51.5 | 49.6 | 51.4 | 48.6 |
| Ce | 93.2 | 95.0 | 138 | 164 | 105 | 101 | 88.1 | 110 | 101 | 103 | 98.3 |
| Pr | 10.6 | 10.8 | 16.6 | 23.8 | 13.4 | 13.0 | 9.9 | 12.4 | 11.3 | 11.6 | 11.2 |
| Nd | 42.5 | 43.7 | 62.6 | 89.3 | 59.5 | 57.6 | 39.4 | 47.4 | 45.5 | 47.5 | 45.7 |
| Sm | 8.39 | 8.65 | 12.0 | 17.6 | 12.8 | 12.3 | 8.27 | 10.1 | 9.60 | 9.83 | 9.60 |
| Eu | 2.78 | 2.83 | 3.76 | 5.38 | 4.17 | 4.03 | 2.72 | 3.39 | 3.13 | 3.19 | 3.14 |
| Gd | 7.08 | 7.23 | 10.7 | 15.7 | 11.0 | 10.6 | 7.12 | 9.20 | 8.27 | 8.93 | 8.53 |
| Tb | 0.89 | 0.94 | 1.50 | 2.30 | 1.40 | 1.35 | 0.93 | 1.30 | 1.08 | 1.14 | 1.10 |
| Dy | 4.84 | 4.99 | 8.00 | 12.2 | 7.55 | 7.31 | 5.05 | 6.60 | 5.94 | 6.23 | 6.05 |
| Ho | 0.79 | 0.81 | 1.30 | 2.10 | 1.24 | 1.19 | 0.82 | 1.10 | 0.97 | 1.01 | 1.00 |
| Er | 1.96 | 2.04 | 3.30 | 5.30 | 3.01 | 2.94 | 2.03 | 2.60 | 2.37 | 2.53 | 2.41 |
| Tm | 0.23 | 0.25 | 0.44 | 0.69 | 0.34 | 0.34 | 0.22 | 0.31 | 0.26 | 0.29 | 0.27 |
| Yb | 1.43 | 1.48 | 2.60 | 4.10 | 2.12 | 2.02 | 1.40 | 1.70 | 1.64 | 1.71 | 1.69 |
| Lu | 0.19 | 0.19 | 0.36 | 0.60 | 0.28 | 0.26 | 0.18 | 0.24 | 0.22 | 0.23 | 0.22 |
| Hf | 5.33 | 5.54 | 5.80 | 6.20 | 6.42 | 6.12 | 3.14 | 3.90 | 3.76 | 4.01 | 3.92 |
| Ta | 3.04 | 2.81 | 4.80 | 5.30 | 3.34 | 2.93 | 2.01 | 2.80 | 2.34 | 2.35 | 2.37 |
| Pb | 3.55 | 3.39 | - | 6.00 | 2.82 | 2.59 | 3.06 | - | 3.68 | 3.64 | 3.59 |
| Th | 4.87 | 5.11 | 7.00 | 7.70 | 3.52 | 3.40 | 4.65 | 6.40 | 5.73 | 6.07 | 5.69 |
| U | 1.29 | 1.29 | 1.40 | 7.70 | 0.97 | 0.91 | 1.08 | 6.40 | 1.28 | 1.27 | 1.26 |

* - ": not detected
*: Analyzed by ActLab

| µg/g | YK18-08 6K#1522R13 Glass LA-ICPMS | YK18-08 6K#1522R16 Glass LA-ICPMS | YK18-08 6K#1522R17 Glass LA-ICPMS | YK19-05S 6K#1542R03 Glass LA-ICPMS | YK19-05S 6K#1542R03 Whole rock * | YK19-05S 6K#1542R05 Glass LA-ICPMS | YK19-05S 6K#1542R06 Glass LA-ICPMS | YK19-05S 6K#1542R09 Glass LA-ICPMS | YK19-05S 6K#1544R04 Glass LA-ICPMS | YK19-05S 6K#1544R04 Whole rock * | YK19-05S 6K#1544R05 Glass LA-ICPMS | YK19-05S 6K#1544R06 Glass LA-ICPMS |
|---|---|---|---|---|---|---|---|---|---|---|---|---|
| Li | 8.06 | 8.53 | 8.42 | 5.54 | | 5.52 | 6.00 | 6.19 | 6.21 | | 6.20 | 6.16 |
| B | 2.83 | 2.77 | 2.94 | 1.60 | | 1.88 | 1.89 | 1.80 | 2.28 | | 2.38 | 2.14 |
| Sc | 21.5 | 19.7 | 20.6 | 22.5 | 24.0 | 22.3 | 22.7 | 23.7 | 22.0 | 22.0 | 22.8 | 23.6 |
| V | 217 | 213 | 209 | 189 | 222 | 188 | 200 | 201 | 203 | 215 | 197 | 191 |
| Cr | 231 | 203 | 203 | 334 | 350 | 317 | 269 | 267 | 292 | 330 | 285 | 273 |
| Co | 44.3 | 47.2 | 46.8 | 42.3 | 49.0 | 42.7 | 42.1 | 41.8 | 44.9 | 47.0 | 43.4 | 42.0 |
| Rb | 28.0 | 30.3 | 29.7 | 14.2 | 14.0 | 14.5 | 17.4 | 17.4 | 17.0 | 17.0 | 16.4 | 16.4 |
| Sr | 930 | 1063 | 1086 | 565 | 487 | 568 | 622 | 643 | 579 | 519 | 595 | 604 |
| Y | 27.0 | 27.9 | 29.6 | 22.8 | 20.0 | 22.4 | 22.5 | 23.7 | 22.9 | 21.0 | 24.0 | 25.1 |
| Zr | 173 | 184 | 194 | 122 | 120 | 122 | 134 | 140 | 123 | 122 | 128 | 132 |
| Nb | 55.7 | 64.2 | 65.7 | 24.0 | 23.0 | 24.0 | 25.1 | 25.9 | 27.0 | 25.0 | 27.3 | 27.4 |
| Cs | 0.36 | 0.41 | 0.40 | 0.18 | | 0.20 | 0.22 | 0.21 | 0.25 | | 0.25 | 0.23 |
| Ba | 514 | 584 | 590 | 255 | 219 | 254 | 292 | 301 | 286 | 259 | 297 | 297 |
| La | 49.3 | 58.1 | 60.9 | 26.8 | 26.1 | 26.6 | 28.6 | 29.8 | 27.8 | 28.0 | 28.8 | 29.5 |
| Ce | 101 | 120 | 122 | 56.6 | 62.8 | 56.5 | 58.8 | 60.4 | 59.8 | 66 | 60.9 | 60.0 |
| Pr | 11.5 | 13.3 | 13.8 | 6.86 | 7.37 | 6.79 | 7.10 | 7.42 | 7.20 | 7.60 | 7.34 | 7.41 |
| Nd | 46.6 | 53.3 | 55.7 | 29.3 | 30.0 | 29.0 | 30.3 | 31.7 | 30.4 | 31.3 | 31.3 | 31.8 |
| Sm | 9.71 | 10.8 | 11.4 | 6.65 | 7.00 | 6.64 | 6.82 | 7.21 | 6.79 | 7.10 | 7.10 | 7.27 |
| Eu | 3.21 | 3.58 | 3.67 | 2.24 | 2.41 | 2.23 | 2.28 | 2.38 | 2.34 | 2.42 | 2.39 | 2.44 |
| Gd | 8.57 | 9.42 | 9.92 | 6.29 | 6.80 | 6.26 | 6.53 | 6.82 | 6.45 | 6.90 | 6.75 | 6.90 |
| Tb | 1.12 | 1.20 | 1.27 | 0.85 | 1.00 | 0.85 | 0.87 | 0.93 | 0.89 | 1.00 | 0.91 | 0.96 |
| Dy | 6.10 | 6.38 | 6.81 | 4.89 | 5.30 | 4.83 | 4.88 | 5.10 | 4.91 | 5.40 | 5.17 | 5.33 |
| Ho | 1.00 | 1.02 | 1.10 | 0.83 | 0.90 | 0.82 | 0.84 | 0.87 | 0.84 | 0.90 | 0.89 | 0.91 |
| Er | 2.46 | 2.47 | 2.63 | 2.12 | 2.30 | 2.13 | 2.10 | 2.22 | 2.10 | 2.30 | 2.27 | 2.32 |
| Tm | 0.28 | 0.28 | 0.30 | 0.26 | 0.28 | 0.26 | 0.26 | 0.26 | 0.26 | 0.29 | 0.28 | 0.27 |
| Yb | 1.70 | 1.67 | 1.75 | 1.57 | 1.70 | 1.57 | 1.52 | 1.60 | 1.58 | 1.70 | 1.66 | 1.71 |
| Lu | 0.22 | 0.21 | 0.22 | 0.21 | 0.23 | 0.21 | 0.20 | 0.22 | 0.21 | 0.22 | 0.23 | 0.23 |
| Hf | 3.95 | 4.08 | 4.36 | 2.95 | 3.10 | 2.95 | 3.20 | 3.39 | 2.95 | 3.00 | 3.12 | 3.18 |
| Ta | 2.40 | 2.63 | 2.77 | 1.08 | 1.30 | 1.10 | 1.16 | 1.23 | 1.21 | 1.40 | 1.23 | 1.24 |
| Pb | 3.71 | 4.38 | 4.29 | 1.67 | | 1.76 | 1.82 | 1.85 | 1.94 | | 1.98 | 1.82 |
| Th | 5.69 | 6.88 | 7.29 | 2.47 | 2.80 | 2.47 | 2.78 | 2.89 | 2.72 | 3.00 | 2.85 | 2.95 |
| U | 1.31 | 1.57 | 1.58 | 0.62 | 2.80 | 0.63 | 0.66 | 0.66 | 0.71 | 3.00 | 0.68 | 0.65 |

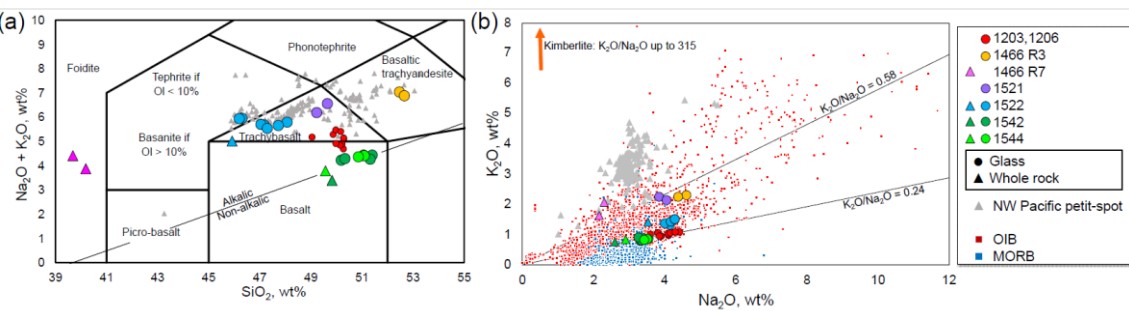

Fig. 4. Relationships between the SiO$_2$ and alkali contents. (a) Total alkali vs. silica diagram using the platform of Le Bas et al. (1986). The dividing line of alkaline and sub-alkaline is from Irvine and Baragar (1971). The data are plotted as the total 100 wt%. The triangles and circles show the whole-rock and quenched-glass compositions, respectively. The compositions of the NW Pacific petit-spots are represented by gray triangles (Hirano and Machida, 2022). The data of the 1203 and 1206 basalts are from Hirano et al. (2019), and those of the 1466R7 basalts are from Mikuni et al. (2022). (b) K$_2$O vs. Na$_2$O diagram. The maximum K$_2$O/Na$_2$O value of kimberlite is from PetDB database (https://search.earthchem.org/). The data of OIB and MORB are compiled from Stracke et al. (2022) as "Expert datasets" in GEOROC database (https://georoc.eu/georoc/new-start.asp).


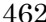

Fig. 5. Selected major-element oxides against MgO. The symbols and compiled data correspond to those in Fig. 3.



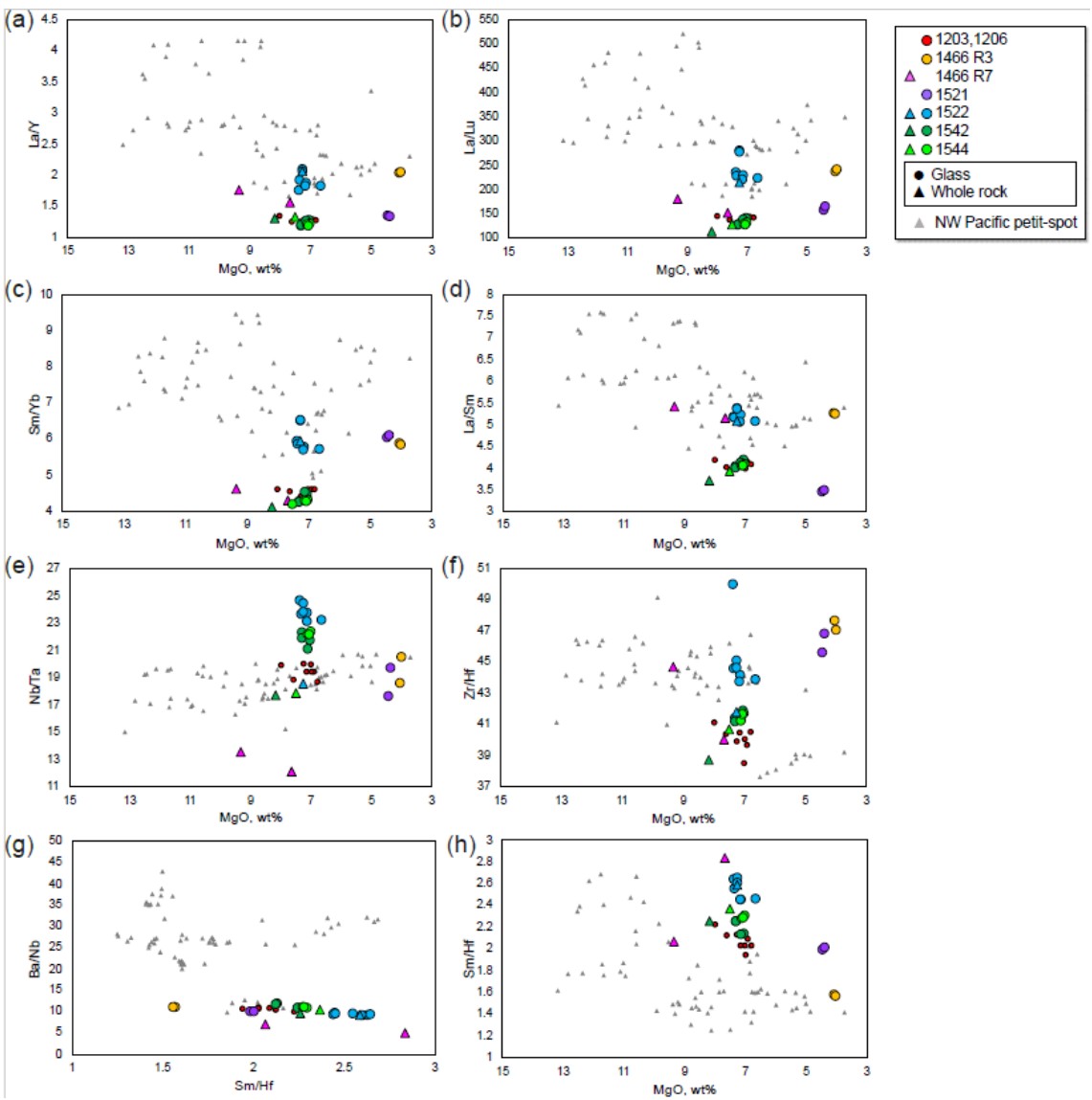


Fig. 6. Selected trace-element ratios against MgO. The symbols and compiled data correspond to those in Fig. 3.

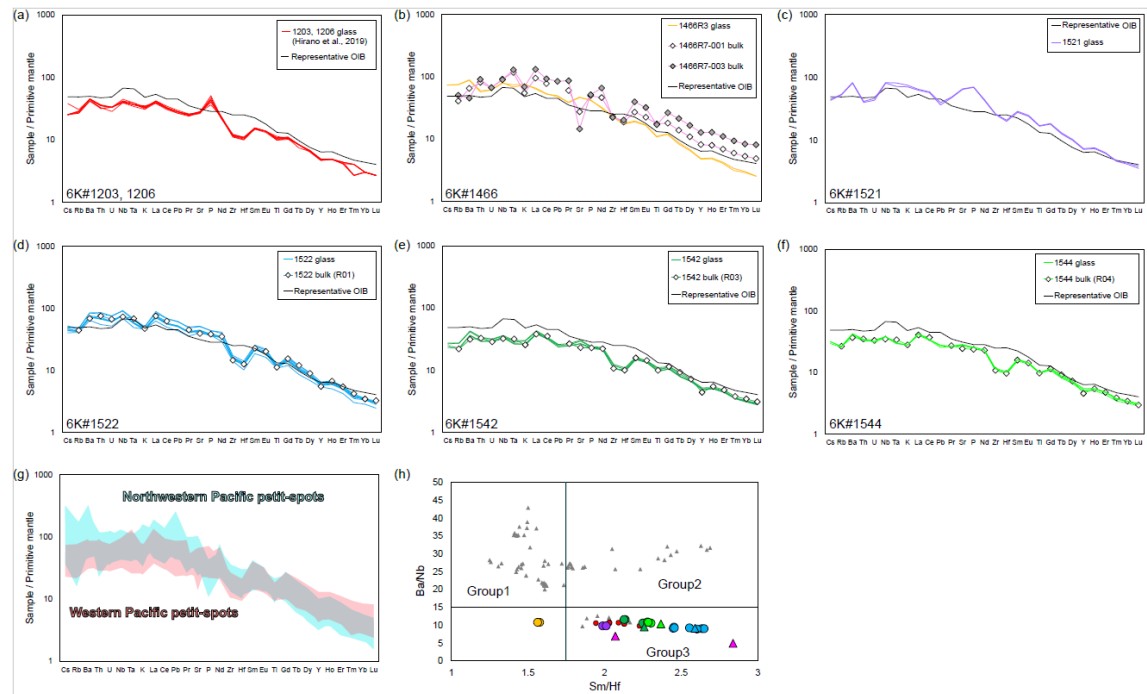


Fig. 7. Primitive mantle (PM, Sun and McDonough, 1989)-normalized trace-element patterns (a)–(g) and element ratios (h). (g) The compositional range of the study samples and NW Pacific petit-spots (Hirano and Machida, 2022). (h) The Ba/Nb and Sm/Hf ratios of the petit-spot basalts to discriminate the three groups after Machida et al. (2015). The data of 1203, 1206 basalts and 1466R7 basalts are from Hirano et al. (2019) and Mikuni et al. (2022), respectively. The symbols and compiled data in the (h) correspond to those in Fig. 3.

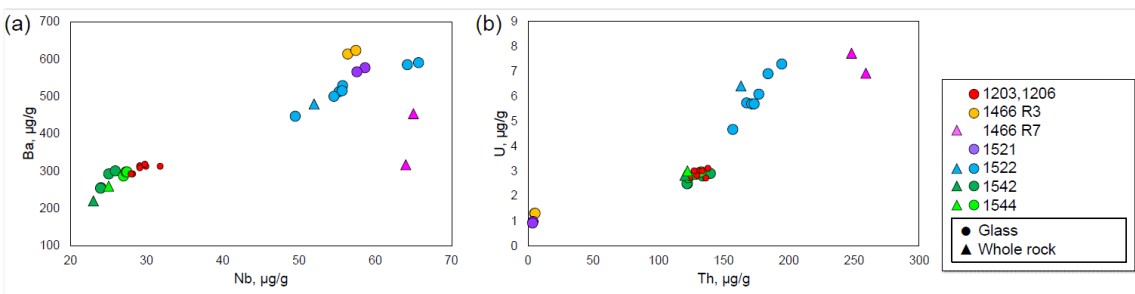

476

Fig. 8. Alteration sensitive elements (Ba and U) vs. insensitive elements (Nb and Th). The symbols and compiled data correspond to those in Fig. 3.

## 5.2 Sr–Nd–Pb isotopic composition

The Sr, Nd, and Pb isotopic compositions of the leached, unleached whole rock, and fresh glasses

in this study (presented in Table 4) were in practically identical ranges of $^{87}Sr/^{86}Sr$ (0.703412–
0.704424), $^{143}Nd/^{144}Nd$ (0.512694–0.512890), $^{206}Pb/^{204}Pb$ (18.6582–18.7778), $^{207}Pb/^{204}Pb$ (15.5086–
15.5749), and $^{208}Pb/^{204}Pb$ (38.6506–38.8041) despite their different locations (Figs. 9a–d, Table 4).
The isotopic compositions of the quenched glass and whole rock were identical, indicating that the
characteristics of the melting source could be obtained through the geochemistry of the young and
fresh volcanic quenched glass. The leached and unleached materials of the same sample also had
similar isotopic ratios, except for the 1466R7-003 basalt, which had a relatively high LOI (6.29 wt%)
(Figs. 9a–d). The Sr–Nd–Pb isotopic three-dimensional (3D) plot is shown in Fig. 9e.

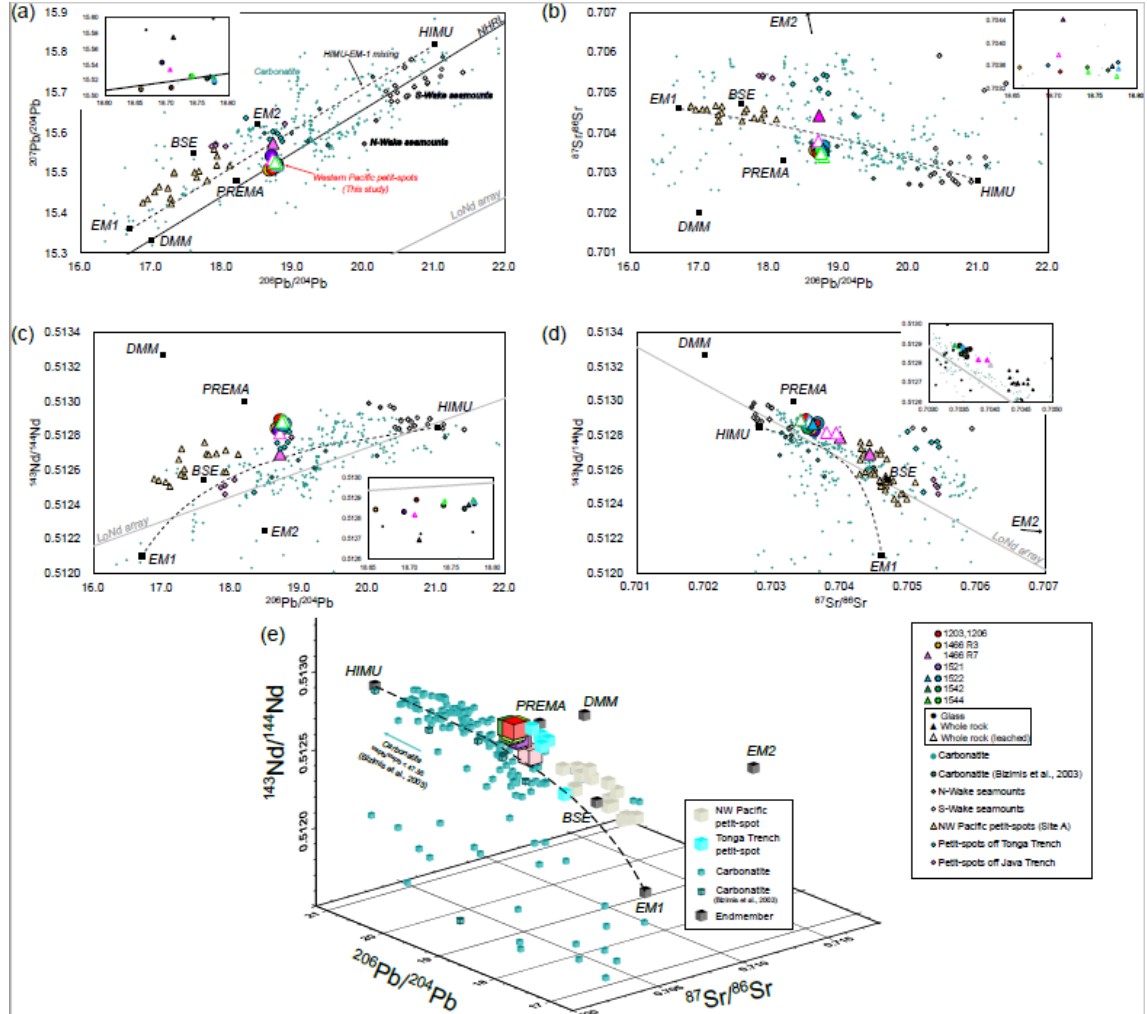

Fig. 9. Sr–Nd–Pb isotopic variations of the petit-spot basalts. The mantle endmembers are derived from a study by

Zindler and Hart (1986). The open triangles in (a)–(d) represent the acid-leached samples. Carbonatite

data were compiled from GEOROC (https://georoc.eu/georoc/new-start.asp) with Bizimis et al. (2003).

Carbonatite data with $^{87}Sr/^{86}Sr$ > 0.706 by GEOROC were eliminated. The northwestern (NW) Pacific

petit-spots and petit-spots off the Tonga Trench are from Hirano and Machida (2022) and Reinhard et al.

(2019), respectively. The petit-spots off the Java trench are from Taneja et al. (2016) and Falloon et al.
(2022). The data of 1203 and 1206 basalts are from Hirano et al. (2019). The data of the Wake seamounts
are from studies by Konovalov and Martynov (1992), Koppers et al. (2003), Konter et al. (2008), Natland
(1976), Smith et al. (1989), and Staudigel et al. (1991). The northern hemisphere reference line (NHRL)
and Low Nd (LoNd) arrays are from studies by Hart (1984) and Hart et al. (1986), respectively. (e) The
three-dimensional (3D) plot of the Sr–Nd–Pb isotopic compositions. The compilation and mantle
endmembers correspond to (a)–(d). The color usages of the plots were the same as (a)–(d).

Table. 4

Sr, Nd, and Pb isotopic compositions of western Pacific petit-spot basalts and measured standards.

| Cruise | Sample name | Sample type | $^{87}Sr/^{86}Sr$ | $^{143}Nd/^{144}Nd$ | $^{206}Pb/^{204}Pb$ | $^{207}Pb/^{204}Pb$ | $^{208}Pb/^{204}Pb$ |
|---|---|---|---|---|---|---|---|
| YK16-01 | 6K#1466 R3-004 | Glass | 0.703568 (06) | 0.512842 (05) | 18.6582 (07) | 15.5086 (06) | 38.6506 (19) |
| YK16-01 | 6K#1466 R7-001 | Whole rock leached | 0.703790 (05) | 0.512817 (07) | 18.7054 (20) | 15.5337 (20) | 38.8041 (50) |
| YK16-01 | 6K#1466 R7-001 | Whole rock unleached | 0.703989 (05) | 0.512790 (06) | | | |
| YK16-01 | 6K#1466 R7-003 | Whole rock leached | 0.703933 (11) | 0.512815 (05) | | | |
| YK16-01 | 6K#1466 R7-003 | Whole rock unleached | 0.704424 (05) | 0.512694 (05) | 18.7107 (06) | 15.5749 (06) | 38.7618 (17) |
| YK18-08 | 6K#1521 R04 | Glass | 0.703605 (05) | 0.512832 (04) | 18.6924 (06) | 15.5428 (06) | 38.7005 (19) |
| YK18-08 | 6K#1522 R01 | Whole rock leached | 0.703544 (05) | 0.512881 (06) | 18.7778 (09) | 15.5209 (08) | 38.7991 (22) |
| YK18-08 | 6K#1522 R01 | Whole rock unleached | 0.703590 (05) | 0.512866 (06) | 18.7705 (07) | 15.5248 (07) | 38.7905 (22) |
| YK18-08 | 6K#1522 R01 | Glass | 0.703656 (06) | 0.512872 (04) | 18.7773 (08) | 15.5178 (07) | 38.7904 (21) |
| YK19-05S | 6K#1542 R03 | Whole rock leached | 0.703412 (07) | 0.512890 (06) | 18.7759 (10) | 15.5244 (11) | 38.7574 (36) |
| YK19-05S | 6K#1542 R05 | Glass | 0.703517 (06) | 0.512847 (04) | 18.7653 (08) | 15.5224 (07) | 38.7345 (19) |
| YK19-05S | 6K#1544 R04 | Whole rock leached | 0.703480 (04) | 0.512883 (05) | 18.7413 (14) | 15.5262 (14) | 38.745 (41) |
| YK19-05S | 6K#1544 R04 | Glass | 0.703568 (05) | 0.512863 (04) | 18.7400 (08) | 15.5253 (09) | 38.7347 (22) |
| YK10-05 | 6K#1206 R04 | Glass | 0.703492 (05) | 0.512890 (04) | 18.7074 (06) | 15.5109 (07) | 38.6970 (19) |
| YK10-05 | 6K#1206 R04 duplicate | Glass | | | 18.7071 (07) | 15.5119 (07) | 38.6950 (18) |

| Type of value | Standared for each isotope | | $^{87}Sr/^{86}Sr$ | $^{143}Nd/^{144}Nd$ | $^{206}Pb/^{204}Pb$ | $^{207}Pb/^{204}Pb$ | $^{208}Pb/^{204}Pb$ |
|---|---|---|---|---|---|---|---|
| Analyzed value | JB-2 | | 0.703721 (05) | 0.513094 (04) | 18.3326 (05) | 15.5453 (06) | 38.2240 (17) |
| Reference value | JB-2 | Sr, Nd: Orihashi et al. (1998), Pb: Tanimizu and Ishikawa (2006) | 0.703709 (29) | 0.513085 (08) | 18.3315 (25) | 15.5460 (21) | 38.2240 (55) |
| Analyzed value | JNdi-1 | (n=2) | | 0.512103 (05) | | | |
| Reference value | JNdi-1 | Wakaki et al. (2007) | | 0.512101 (11) | | | |
| Analyzed value | SRM987 | (n=2) | 0.710239 (05) | | | | |
| Reference value | SRM987 | Weis et al. (2006) | 0.710254 (02) | | | | |
| Analyzed value | SRM981 | | | | 16.9303 (05) | 15.4828 (06) | 36.6710 (16) |
| Reference value | SRM981 | Tanimizu and Ishikawa (2006) | | | 16.9308 (10) | 15.4839 (11) | 36.6743 (30) |

Errors shown in parentheses represent 2σ and apply to the last two digits.

## 5.3 Age determination and estimation


The $^{40}Ar/^{39}Ar$ ages were determined for two samples (1466R6-001 and 1522R01) (Fig. 10a,
Table S4). The secondary material (e.g., alteration products) plausibly causes the recoil loss and
redistribution of Ar during irradiation of samples, particularly fine-grained groundmass separates of
submarine basalt (Koppers et al., 2000). This effect is negligible for $^{40}Ar/^{39}Ar$ dating samples in this
study because the total K/Ca ratios estimated using the irradiated $^{39}Ar_K/^{37}Ar_{Ca}$ ratio (0.089 for 1466R6,
0.080 for 1522R01; Table S4) are mostly correspond to the bulk K/Ca ratios calculated using the major
element compositions of Table 2 (0.088 for 1466R6-001, 0.076 for 1522R01). This is supported by
the rock descriptions recognized no secondary materials of crystalline $^{40}Ar/^{39}Ar$ specimens. The
1466R6-001 sample had a plateau age of 3.03 ± 0.18 Ma in seven fractions comprising 94.1% released
$^{39}$Ar. However, the plateau age was recognized as apparently old, owing to excess $^{40}$Ar, as indicated
by the initial $^{40}$Ar/$^{36}$Ar ratio of 325 ± 15, which exceeded the atmospheric ratio (296.0; Nier, 1950) in
the inverse isochron. The inverse isochron age of 2.56 ± 0.34 Ma showed the best age estimate for the
1466R6-001 basalt (Fig. 10a). The 1522R01 sample released almost no radiogenic daughter nuclide
of $^{40}$Ar in the K–Ar age system (Fig. 10a).
The ranges of eruption age were estimated for all the samples using the average thickness (n =
20) of ferromanganese crust and palagonite rind (hydrated quenched glass) with their
deposition/formation rates on the seafloor (ferromanganese crust, 1–10 mm/Myr; Hein et al., 1999;
palagonite, 0.03–0.3 mm/Myr; Moore et al., 1985) (Fig. 10b). Using this approach, the western Pacific
petit-spots were expected to have erupted later than ca. 9 Ma. The ranges of eruption age estimated
from palagonite rind did not overlap with those from ferromanganese crust showing older durations,
although they had general correlations (Fig. 10b). The $^{40}$Ar/$^{39}$Ar ages of two samples and the U-Pb
age of zircon in the 1203 and 1206 peperites (Hirano et al., 2019) were overlaid within these ranges.

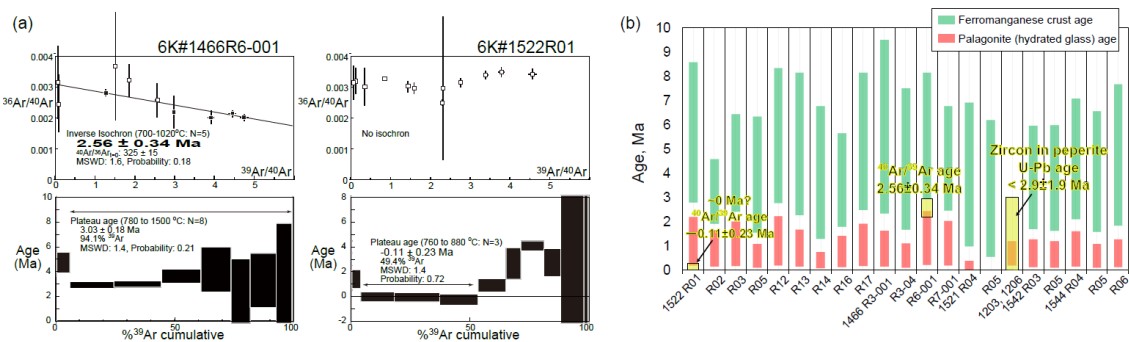

Fig. 10. Geochronological data. (a) The $^{40}$Ar/$^{39}$Ar ages of the 6K#1466R6-001 and 6K#1522R01 basalts. The errors
show a 2-sigma confidence level. (b) Estimated relative ages using the thickness of ferromanganese crust
(green bands) and palagonite (hydrated quenched-glass rind; red bands) covered with petit-spot basalts.
These values were estimated using the average for each sample (n = 20). The U-Pb age of zircon in the
6K#1203 and 1206 peperites are from Hirano et al. (2019).

**6 Discussion**

**6.1 Eruptive setting of western Pacific petit-spots**

In this study, two crystalline petit-spot basalts were subjected to $^{40}$Ar/$^{39}$Ar dating. A previously
investigated petit-spot knoll in this region (examined during the 6K#1203 and #1206 dives) was dated
at "younger than 3 Ma" through the U–Pb dating of eight zircons in peperites (Fig. 10b) (Hirano et al.,
2019). The results revealed that the silica-undersaturated vesicular basalt of 1466R6-001, hosting
ultramafic xenoliths (Mikuni et al., 2022), exhibited a $^{40}Ar/^{39}Ar$ age of 2.56 ± 0.34 Ma (Fig. 10). On
the contrary, the fresh vesicular basalt of 1522R01, which erupted at the foot of the 100-Ma Takuyo-
Daigo seamount (Fig. 2) (Nozaki et al., 2016), did not exhibit radiogenic $^{40}Ar$ indicating its young age
(~0 Ma) (Fig. 10). The ranges of eruption ages were estimated using the average thickness of
ferromanganese crust and palagonite rind (seawater-hydrated quenched glass) with their
deposition/formation rates on the seafloor. The $^{40}Ar/^{39}Ar$ and zircon U–Pb ages were within these
ranges (Fig. 10). The petit-spot volcanic field is surrounded by Cretaceous seamounts (Koppers et al.,
2003) and irregular Paleogene volcanoes (Aftabuzzaman et al., 2021; Hirano et al., 2021). However,
no zero-aged hotspots were observed in this region, and the P-wave tomographic image of the surface
to the core–mantle boundary of the study area did not exhibit a plume-like low-velocity zone (Fig. 1c;
Lu et al., 2019). Furthermore, the MORB-like to more depleted noble-gas isotopic compositions of
the petit-spot knoll (investigated by 6K#1203 and #1206 dives) suggested its upper mantle origin
(Yamamoto et al., 2018). Along with the outer-rise bulge in front of the Mariana Trench detected
through a positive gravitational anomaly (Hirano et al., 2019), these data suggest that the western
Pacific petit-spot volcanoes could have erupted at ~0–3 Ma owing to the flexure of the subducting
Pacific Plate into the Mariana and Ogasawara Trenches.
The petit-spot basalts from the 6K#1542 and #1544 dives could have originated from the same
eruptive source based on their similar petrographic and geochemical features despite a distance of ~6.8
km between both (Figs. 3d, 4, 5, 6, 7, 8, and 9). Contrarily, in terms of their petrography and
geochemistry, the basalts from the 6K#1466 dive are distinguished between the samples from the lava
flows on the abyssal plain (1466R3-001 and 1466R3-004 samples) and the samples from the knoll site
(1466R6-001, 1466R7-001, and 1466R7-003 samples). The 1466R3 basalts were collected at a lava
outcrop 600 m south of the knoll, and the 1466R6 and 1466R7 samples were collected on the western
slope of the knoll (Fig. 3a). The 1466R3 series are glassy with a high $SiO_2$ content (50.6–51.6 wt%),
including minor plagioclase and fewer vesicles (Figs. 3a and 4a). However, the 1466R6–R7 series
exhibited silica-undersaturated compositions ($SiO_2$ = 39.3–39.4 wt%) and high vesicularities (20–40
vol.%) (Figs. 3b and 4a). Combining these observations with the differences in MgO contents and
trace element compositions, the 1466R3 and 1466R6–R7 basalts are implied to have different parental
magmas (Figs. 6 and 7b). Generally, vesicular samples (1203, 1206, 1466R7, 1522, 1542, and 1544
basalts) are relatively primary (i.e., MgO > 6.63 wt%), whereas nonvesicular samples (1466R3 and
1521 basalts) are evolved (i.e., MgO < 4.43 wt%). This correlates with the compositions of olivine
microphenocrysts in the low forsterite content (Fo# = 100 × Mg/[Mg+Fe$^{2+}$]$_{cation}$) of olivine in evolved
basalts and the high Fo# of olivine in the relatively primary basalts (Figs. S1a–c).
The CI chondrite-normalized REE ratios of these samples are within those of OIBs, and the
REE patterns exhibit HREE-depleted patterns (Fig. S3). However, among the western Pacific petit-
spots, each volcano shows distinct REE and trace element ratios (i.e., parental magmas) (Figs. 6 and
S3). Considering the absence of correlation between MgO and the trace element ratios, it is suggested
that each volcano could have originated from isolated sources (i.e., melt ponds) with varying chemical
compositions and degrees of melting (Fig.6). On the contrary, the radiogenic Sr, Nd, and Pb isotopic
ratios of the samples are nearly identical, indicating equivalent components in the source (Fig. 9).
In summarily, (1) the western Pacific petit-spot volcanoes erupted at ~0–3 Ma owing to the plate
flexure related to the subduction of the Pacific Plate into the Mariana Trench (Figs. 1 and 2). (2) The
1542 and 1544 samples originated during the same magmatic event (Fig. 3d). However, the basalts
from the 6K#1466 dive were divided into two parental magmas (1466R3 and 1466R6–R7 basalts)
(Fig. 3a). (3) Each volcano originated from an isolated source and/or ascending processes, as indicated
by independent trace element ratios. Despite this, the geochemical components involved in the source
were similar among the western Pacific petit-spot volcanoes due to the nearly identical Sr, Nd, and Pb
isotopic compositions (Figs. 6 and 9). The variation in trace element compositions among the
volcanoes is plausibly attributed to the degree of contribution of carbonatite flux and/or the recycled
crustal component to the source, as discussed below.

**6.2 Petit-spot magma composition and its evaluation**

Post-eruption alteration in seawater may have affected the chemical composition of oceanic
basalts. Thus, various approaches, including petrographic observation, geochemical investigation, and
acid leaching, have been employed to evaluate the primary features and the removal of this effect for
isotopic analysis (Hanano et al., 2009; Melson et al., 1968; Miyashiro et al., 1971; Nobre Silva et al.,
2009; Resing and Sansone, 1999; Staudigel and Hart, 1983; Zakharov et al., 2021). The study samples
exhibit whole-rock LOI of <1.72 wt%, except for two relatively altered samples, 1466R7-001 (LOI =
2.68 wt%) and R7-003 (LOI = 6.29 wt%) basalts. Pristine quenched glasses are preserved in most of
the samples, excluding three exceptional samples (1466R6-001, R7-001, and R7-003 basalts). Positive
correlations exist between the alteration-insensitive (e.g., Nb and Th) and -sensitive (e.g., Ba and U)
incompatible elements, indicating that the effect of seawater alteration was not extensive, except for
the 1466R7-001 and R7-003 basalts (Fig. 8). Despite originating from different volcanic edifices, the
positive correlation of all the study samples is attributed to the chemical similarity of source
compositions for certain elements (i.e., the Ba/Nb and U/Th ratios are nearly constant among the
samples) as well as the Sr, Nd, and Pb isotopic compositions (Fig. 9). These findings demonstrate that
most of the petit-spot basalts were largely unaffected by seawater alteration, with a few exceptions,
i.e., 1466R7-001 and R7-003 basalts.
The MgO (4–9 wt%), Ni (<263 ppm), and Cr (<350 ppm) contents in the samples are lower than
the expected values of primary mantle-derived melt (MgO >10 wt%, Ni >400 ppm, Cr >1000 ppm;
Frey et al., 1978). Similarly, the Mg# ($100 \times Mg/[Fe^{2+} + Mg]_{molar}$) values range from 41 to 57 (Table
2) against the primary basaltic melt, which is equilibrated with the upper mantle (Mg# = 66–75; Irving
and Green, 1976). No phenocrysts were observed (only microphenocryst), despite such differentiated
compositions as well as most of the NW Pacific petit-spot basalts. This suggests that the western
Pacific petit-spots experienced crystal fractionation in the lithosphere as well as the case in the NW
Pacific petit-spot (Machida et al., 2017; Valentine and Hirano, 2010; Hirano, 2011; Yamamoto et al.,
2014). Consequently, calculating the primary composition of the petit-spot basalts using the mineral
modal composition on the thin section was not possible. However, the major element trends of the
samples indicate the crystal fractionation of the same phases. Negative trends of the $Al_2O_3$ content and
the positive trends in CaO and $CaO/Al_2O_3$ content with decreasing MgO indicate the occurrence of
olivine, spinel, and clinopyroxene fractionation (Figs. 5c, e, and g). The absence of visible correlations
of $K_2O$, $Na_2O$, $SiO_2$, and $TiO_2$ contents against MgO suggests insignificant fractionation of plagioclase
and Fe–Ti oxides. The Fe–Ti oxides as minor phases in the groundmasses and plagioclases were only
observed in the most differentiated 1466R3-001 and R3-004 basalts (Figs. 3, 5a, b, d, and h). However,
these major elemental trends should be interpreted as apparent because each petit-spot volcano
originated from an isolated parental magma with a different chemical composition or degree of partial
melting, as discussed above.

The melting source of alkali basalts can be determined more effectively by examining their trace

element composition rather than major elements (Hofmann, 2003; Machida et al., 2014, 2015). Trace
element composition of magma, however, could be modified by crustal and/or mantle assimilation and
fractionation of specific minerals. The relatively primitive basalts (1203, 1206, 1466R6, R7, 1522,
1542, and 1544 samples) contained xenocrystic olivines and partly ultramafic xenoliths, suggesting a
rapid magma ascent (Hirano et al., 2019; Mikuni et al., 2022; Fig. S4). However, since the stagnation
of ascending petit-spot magma could lead to the formation of fertile peridotite and pyroxene-rich veins
in the middle to lower depths of the lithosphere (Mikuni et al., 2022; Pilet et al., 2016), the chemical
composition of the petit-spot magma could be modified through assimilation with ambient lithospheric
peridotite. According to Hirano and Machida (2022), ascending silica-undersaturated melt would
predominantly consume orthopyroxene (±spinel) and result in a more silicic composition with Zr and
Hf depletion. This is due to the relatively higher Zr–Hf partition of orthopyroxene than compared to
other trace elements (Pilet et al., 2008; Shaw, 1999; Tamura et al., 2019). The orthopyroxenes of fertile
pyroxenites and lherzolite xenoliths metasomatized by petit-spot melts exhibit Zr and Hf enrichment
(Mikuni et al., 2022; Fig. S5). If this silica-enrichment (i.e., melt–rock interaction) was significant, a
positive correlation between $SiO_2$ and Sm/Hf is expected as a mantle assimilation trend. However, the
samples exhibited a negative correlation, similar to those of the NW Pacific petit-spots (Hirano and
Machida, 2022) (Fig. S2). Considering the relation between the Sm and Hf partition coefficients of
clinopyroxene (i.e., $D^{Hf} < D^{Sm}$; McKenzie and O'Nions, 1991; Kelemen et al., 2003), we suggest that
the negative correlation between the Sm/Hf and $SiO_2$ in the petit-spot basalts probably reflects the
crystal fractionation of clinopyroxene rather than mantle assimilation. The Ba/Nb ratios of the samples
are nearly constant and do not correlate with the MgO and $SiO_2$ contents (Figs. 6g and S2g). The lack
of correlation between other trace element ratios, excluding Sm/Hf and Ba/Nb (i.e., La/Y, La/Lu,
Sm/Yb, La/Sm, Nb/Ta, Zr/Hf), and the MgO concentration suggests that crystal fractionation may not
have been involved in those of the incipient melt (Fig. 6). However, independently tracking the
evolution of the trace element composition for each volcano is challenging, given that each volcano
originated from isolated sources. Thus, considering the observations above, the fresh and zero-aged
1522 basalts (having the highest Sm/Hf ratios and lowest $SiO_2$ contents among the fresh samples and
higher MgO contents) were selected for further analysis with geochemical modeling. Given that the
1522 samples had MgO in the range of 6.63–7.36 wt%, olivine was expected to be the dominant phase
of crystal fractionation (Asimow and Langmuir, 2003; Helz and Thornber, 1987; Herzberg, 2006). By
applying the olivine maximum fractionation model (Takahashi et al., 1986; Tatsumi et al., 1983) to
test two samples, it was noted that 7–9% olivine addition was required to achieve the olivine
composition corresponding to "Mantle olivine array" in the NiO and Fo# spaces (Figs. S6a, b). The
calculated primary trace element contents did not considerably differ from those of the analytical
compositions (Table S5 and Fig. S6). Thus, the 1522 basalts were assumed to be the most primary
petit-spot basalt samples and were used to evaluate the geochemical modeling results.

**6.3 Melting source of western Pacific petit-spots**

The depletions observed in specific elements (e.g., Ta, Zr, Hf, and Ti) in the petit-spot basalts
potentially demonstrate the involvement of carbonatitic materials in conjunction with a large amount
of $CO_2$ and lower Mg isotopic ratio than that of the normal mantle (Bizimis et al., 2003; Dasgupta et
al., 2009; Hirano and Machida, 2022; Hoernle et al., 2002; Liu et al., 2020; Okumura and Hirano,
2013). Other oceanic lavas originating from the asthenosphere (e.g., Hawaiian rejuvenated lavas and
North Arch volcanoes) exhibited characteristic trace element signatures (i.e., Zr and Hf depletion)
similar to those of petit-spot lavas. This implies that their melting sources were involved with
carbonatitic materials with or without plume-derived components (Fig. S7; Borisova and Tilhac, 2021;
Clague and Frey, 1982; Clague et al., 1990; Dixon et al., 2008; Yang et al., 2003). Additionally, the
involvement of recycled crustal components was inferred from the geochemical features of the petit-
spot basalts, and the upper mantle was revealed to be heterogeneous (Liu et al., 2020; Machida et al.,
2009, 2015). Such a scenario of the source for petit-spot magma aligns with the previously suggested
petrogenesis of alkaline rocks explained by the addition of $CO_2$-rich components and/or recycled
crustal materials with or without sediment to the mantle (e.g., Dasgupta et al. 2007; Hofmann, 1997).
Conversely, the melting of an amphibole-rich metasomatic vein explains the major and trace element
composition of alkali basalts (Pilet et al., 2008; Pilet, 2015). However, the experimentally produced
melts exhibit Pb depletion and a positive Nb-Ti anomaly in the PM-normalized trace element patterns
(Fig. S8), which is inconsistent with the petit-spot basalts (Fig. 7). Moreover, Juriček and Keppler
(2023) demonstrated that amphibole dehydration is not the cause for the oceanic LAB through high-
pressure experiments under the realistic conditions. The fertile pyroxenitic xenoliths and pyroxene
xenocrysts in the 1466R6 and R7 basalts, originating from the metasomatic vein related to prior petit-
spot magmatism, had neither amphiboles nor other hydrous minerals (Mikuni et al., 2022).

To explore the involvement of carbonatitic and crustal components in petit-spot melts, a partial
melting model of the heterogeneous mantle is presented. The involvement of carbonatitic fluids and
recycled materials in the genesis of petit-spot melts has been suggested, and the open-system model
with carbonatite influx from the outer system was employed using "OSM-4" by Ozawa (2001), and
by referring the parameters by Borisova and Tilhac (2021). This model is based on the mass
conservation equations of one-dimensional steady-state melting. In this study, the model asset the
critical melt fraction ($\alpha_c$; mass fraction of melt when melt separation begins = melt connectivity
threshold) at 0.005 or 0.01. The system opens to fluxing at a constant melt-separation rate ($\gamma$) when
the system reaches the $\alpha_c$. The final trapped melt fraction ($\alpha_f$; mass fraction of melt trapped in the
residue) was fixed at ~0 (it was calculated as $10^{-6}$ owing to mass balance). We calculated the trace
element composition of partial melts at various degrees of melting ($F$) as well as a few rates of influx
($\beta$) and melt separation ($\gamma$). We assumed a primitive mantle (PM) source as the lherzolite with or
without a normal (N)-MORB source as the recycled oceanic crust (Sun and McDonough, 1989), such
as pyroxenite and eclogite. The recycled crust (N-MORB component) was mixed in the source as
compositional heterogeneity calculated as "0.05N-MORB + 0.95PM" for trace element concentration.
The mineral phases and their proportions considered were derived only from garnet lherzolite (i.e.,
olivine, orthopyroxene, clinopyroxene, and garnet). The mineral mode of garnet lherzolite (olivine
55%, orthopyroxene 20%, clinopyroxene 15%, and garnet 10%) and the melting reaction mode
(olivine 8%, orthopyroxene −19%, clinopyroxene 81%, and garnet 30%) are based on studies by
Johnson et al. (1990) and Walter (1998), respectively. The proportion of olivine and garnet was also
changed to assess the effect of the garnet modal ratio on the produced melt composition. In this
situation, the clinopyroxene is consumed at a degree of partial melting of ~ 19%; hence, the system
was calculated up to 18% partial melting. The carbonatite melt used in this model as a influx is
"average carbonatite" from a study by Bizimis et al. (2003). The partition coefficient of trace elements
is generally based on a study by McKenzie and O'Nions (1991, 1995), excluding Ti for clinopyroxene
and garnet (Kelemen et al., 2003). The variables of $\beta$ (influx rate) and $\gamma$ (melt-separation rate) were
changed during the modeling within the mass balance ($\gamma \leqq \beta + 1$). The modeled melts were outputted
as "total melt," considering the instantaneous and accumulated melts. For the carbonatite composition,
the value of "average carbonatite" from Bizimis et al. (2003) is applied because the chemical

composition of carbonatite is largely diverse, and this value is recommended for geochemical modeling (Bizimis et al., 2003). The parameters are detained in Table S6. Consequently, partial melting of garnet lherzolite with a 10% carbonatite influx to a given mass of source (i.e., garnet lherzolite) can provide a rough explanation of the trace element pattern of petit-spot basalts (Figs. 11a–e). The most plausible for petit-spot magma generation involves the presence of a 5% crustal component in the source (Figs. 11b and d). In addition, having slightly less garnet in the lherzolite source than the modal ratio of Johnson et al. (1990) offers a better fit for petit-spot characteristics (Fig. 11b). In both scenarios, incorporating a crustal component in the source produces more plausible outcomes (Figs. 11a–d). The higher carbonatite influx ($\beta$ =1.0) could not explain the trace element composition of the petit-spot basalts (Fig. 11f). A melt connectivity threshold ($\alpha_c$) of 0.01 is considered plausible, as higher connectivity of melt (i.e., lower $\alpha_c$ value) leads to enrichment of LILEs and LREEs (Fig. 11g). The results also indicate that the melt-separation ratio has no significant impact on the trace element composition of the calculated melts (Figs. 11d and e). Thereafter, we concluded that the partial melting of ~5% crustal component-bearing garnet lherzolite with ~10% carbonatite flux to a given mass of the source plausibly explains the melting source of petit-spot volcanoes (Figs. 11b and d). Assuming that the trace element composition of 1203, 1206, 1542, and 1544 basalts are also primitive, they could be explained by the partial melting of garnet lherzolite with 5% crustal component and lower carbonatite influx rate ($\beta$ = 0.03) (Fig. S9). Actually, the 1203, 1206, 1542, and 1544 basalts exhibited similar MgO contents and Mg# to those of the 1522 basalts (Fig. 4 and Table 2). These results provide quantitative evidence regarding petit-spots' petrogenesis, i.e., the contribution of carbonatite melt and recycled oceanic crust.

Although the melting source included small proportions of carbonatite melt and crustal components, these components could have contributed to isotopic composition owing to their abundant incompatible elements, as opposed to the ambient mantle. Determination of the Sr, Nd, and Pb isotopic compositions indicated that they had geochemically identical prevalent mantle (PREMA)-like sources (Fig. 9). Contrary to those of NW Pacific petit-spots, which exhibit EM-1 isotopic composition (Machida et al., 2009; Liu et al., 2020), the samples herein did not align with any mantle isotopic endmembers (i.e., depleted MORB mantle (DMM); EM-1 and EM-2; and HIMU; Fig. 9). In the Pb isotopic space, the present samples did not correlate with those of the neighboring HIMU-like Cretaceous seamounts (Fig. 9a) (N-Wake, S-Wake seamounts; Konter et al., 2008; Koppers et al., 2003; Natland, 1976; Smith et al., 1989; Staudigel et al., 1991). For the melting source of the NW Pacific petit-spot basalts, the involvement of the eclogite/pyroxenite endmember as recycled oceanic crust and the carbonated endmember was suggested. This suggestion was based on the major and trace elements and the Mg, Sr, Nd, and Pb isotopic compositions with Mg diffusion modeling (Liu et al., 2020). The higher FeO/MnO ratios observed in the present melts (65.9–78.0), compared to those of partial melts originating from peridotite (50–60), are attributed to the presence of recycled pyroxenite

(Herzberg, 2011), potentially contributing to crustal components in the melting source. However, the
western Pacific petit-spots in this study uniformly displayed a PREMA-like isotopic signature without
extreme endmember contributions, as described previously (Fig. 9). Such isotopic compositions with
the world's petit-spots can be possibly explained by the diverse mixing proportion of HIMU and EM-
1 components (Fig. 9e). The isotopic compositions of the NW Pacific petit-spots (off the Japan Trench),
Samoan petit-spots (off the Tonga Trench), petit-spot dikes in Christmas Island (off the Java trench),
and western Pacific petit-spots (off the Mariana Trench in this study) are roughly along the HIMU–
EM-1 mixing line (Fig. 9e). Furthermore, the isotopic compositions of global carbonatites can
generally be explained by the mixing of HIMU and EM-1 (Bell and Tilton, 2002; Hoernle et al., 2002;
Hulett et al., 2016). The contributions of the carbonated material/carbonatite and crustal components
to the melting source were suggested in relation to the origin of HIMU and EM-1 (Collerson et al.,
2010; Hanyu et al., 2011; Wang et al., 2018; Weiss et al., 2016; Workman et al., 2004; Zindler and
Hart, 1986). However, the determination of EM-1 and HIMU components as carbonated components
and recycled crust, respectively, is challenging due to the varied perspectives on each tectonic setting
for the mantle endmember. The variability of global carbonatite isotopic compositions poses
challenges in determining their representative isotope ratios (Fig. 9). Despite these challenges
hindering a quantitative isotopic mixing model, the HIMU-EM-1-like trend observed in global petit-
spot volcanoes suggests the involvement of carbonatitic and recycled crustal materials. In conclusion,
the mass balance models applied to trace elements and the isotopic variations in the petit-spot
volcanoes confirmed the contribution of carbonatite melt and the recycled oceanic crust to the melting
source of the western Pacific petit-spots (Fig. 12). Experimental studies have revealed the diverse
petrogenesis scenarios of carbonatite and carbonatitic alkali-rich magma under high pressures
(Dasgupta et al., 2006; Ghosh et al., 2009). The geochemistry of petit-spot basalts including Mg
isotopes suggested that the conceivable origin of carbonatite related to the petit-spot melt is subducted
"carbonated" pelite, pyroxenite/eclogite, or peridotite stored as diamond or metal carbide in the
reduced lower portion of the upper mantle (Liu et al., 2020; Rohrbach et al., 2007). For instance,
subducted carbonated pelite would melt under high pressure (>8 GPa) through oxidation at the redox
boundary where the iron-wüstite (IW) buffer changes to the quartz–fayalite–magnetite (QFM) buffer
(i.e., redox melting; Grassi and Schmidt, 2011). Chen et al. (2022) demonstrated that the alkali-rich
carbonatite melt could occur at a pressure exceeding 6 GPa, particularly exhibiting K-rich and Na-rich
carbonatites under 6–12 and >12 GPa, respectively. This pressure-dependent alkalinity of the resulting
carbonatite melts could potentially account for the differences between potassic NW Pacific petit-spot
lavas and present sodic petit-spot lavas (Fig. 4b). On the other hand, an experimental study highlighted
the presence of a carbonate-rich layer in the LAB owing to the horizontally spread carbonate from
around the wedge mantle rather than upwelling from the deep mantle (Hammouda et al., 2020). Several
high pressure–temperature experiments and modeling revealed that the chemical composition of
intraplate magmas originating from the upper mantle depends on their original depth. Specifically, the
carbonatitic melt can be generated beneath thick cratonic lithosphere (~250–200 km), kimberlitic melt
could be produced at >120 km in depth, and alkali basalt could occur at 100–60-km depth by the
partial melting of "original" $CO_2$ and $H_2O$-bearing mantle (Massuyeau et al., 2021). This depth-
dependent variation in composition, i.e., K-rich kimberlite to alkali basalt, may provide an explanation
for the geochemical gap between K-rich NW Pacific petit-spots and K-poor western Pacific petit-spots
(Fig. 4b). Although the multiple origins of carbonatite are merely suggested and remain unclear,
carbon-rich components play a key role in the partial melting of mantle at the LAB (Sifré et al., 2014),
constituting the source of petit-spot magma.

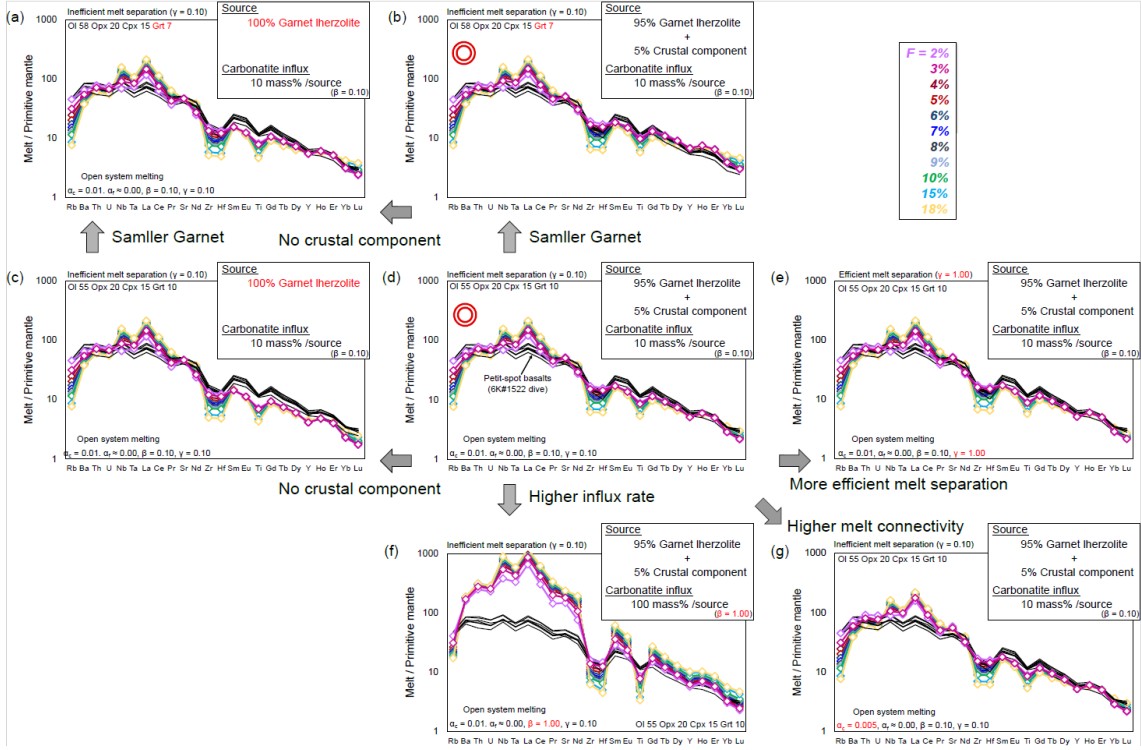


Fig. 11. Geochemical modeling for the primitive mantle (PM)-normalized trace-element pattern. The calculated
hypothetical melts are a production of carbonatite influx melting of garnet lherzolite with or without 5%
crustal component. Detailed information of the parameters is described in Section 6-3 and Table S6. $F$ is
the degree of melting (%). The trace-element composition of the western Pacific petit-spot basalts from
the 6K#1522 dive is shown as black lines for comparison. The PM composition of lherzolite and the N-
MORB composition of recycled crust were based on a study by Sun and McDonough (1989). The influx
carbonatite is the "average carbonatite" of a study by Bizimis et al. (2003). The parameters used in the
open-system melting models were as follows: $a_c$ is a critical melt fraction, $a_f$ is a final trapped melt
fraction, β is a melt influx rate, and γ is a melt-separation rate. Model results are compared by varying
each parameter, i.e., garnet modal ratio and presence of crustal material (a–d), melt-separation rate (d and
e), carbonatite influx rate (d and f), and critical melt fraction (d and g). Each figure is expressed based on

the difference from the condition in (d).

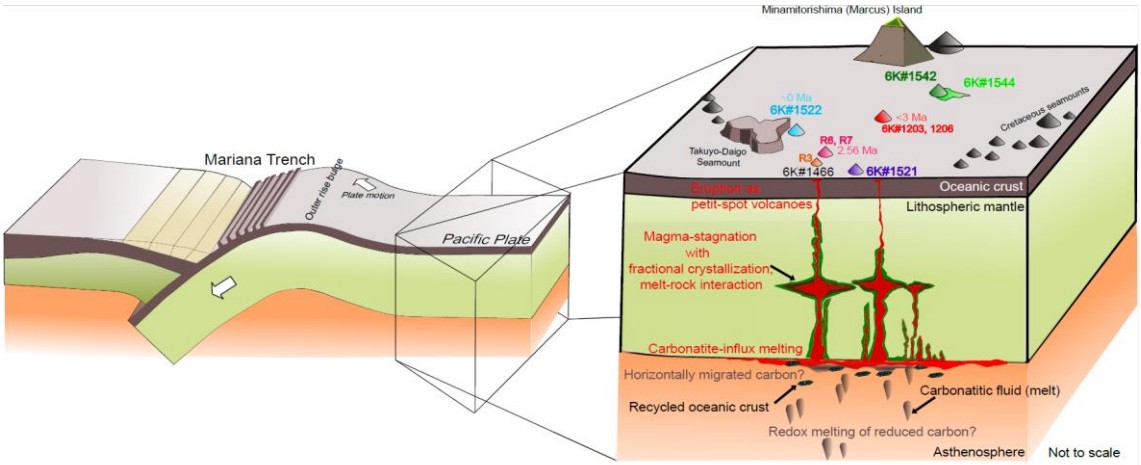

Fig. 12. Schematic illustration of the magmatic processes of the western Pacific petit-spot volcanoes.

Carbonatitic melt and recycled oceanic crust potentially induce partial melting of asthenospheric mantle

beneath the western Pacific region. Carbonatitic melt might have originated from a carbon-rich

component horizontally migrated from a subduction zone (Hammouda et al., 2020), or a redox melting

of reduced carbon in the deep mantle (Chen et al., 2022; Grassi and Schmidt, 2011; Rohrbach et al., 2007).

Petit-spot magma stagnated in the lithosphere with fractional crystallization and melt-rock interaction

(Mikuni et al., 2022), and they have erupted at ~0–3 Ma.


**7 Conclusion**

The occurrence of petit-spot volcanism supports partial melting at the LAB, carrying significant

implications for the characteristics of this geophysical discontinuity. Numerous instances of petit-spot
magmatism occurred on the western Pacific Plate at ~0–3 Ma, originating from similar PREMA-like
melting sources based on $^{40}Ar/^{39}Ar$ dating and the Sr, Nd, and Pb isotopic compositions. The mass
balance-based open-system modeling for trace elements revealed that the western Pacific petit-spot
magma was generated by the partial melting of a small amount (5%) of oceanic crust-bearing garnet
lherzolite with 3%–10% carbonatite influx to a given mass of the source. The isotopic compositions
of Sr, Nd, and Pb of the study samples, in conjunction with those of the NW Pacific petit-spots, petit-
spots off the Tonga and Java Trenches, could be explained by mixing the EM-1-like and HIMU-like
components, contributing to the subducted carbonated/crustal materials. The tectonic-induced
magmatism, such as a petit-spot, may follow a similar melting mechanism.

**Authorship contributions**

K. Mikuni and N. Hirano conceived the project and performed all experiments. S. Machida and Y. Kato contributed the Sr, Nd, and Pb isotopic analysis using TIMS and MC-ICP-MS. H. Sumino contributed the $^{40}Ar/^{39}Ar$ dating. N. Akizawa, A. Tamura, and T. Morishita helped and performed EPMA and LA-ICP-MS analyses. S. Machida and N. Hirano conducted the research cruises to gain the rock samples. All authors interpreted the data and wrote the manuscript with comments and improvements.

**Competing Interests**

The authors declare that they have no conflict of interest.

**Data availability**

The data newly analyzed in this study and results of geochemical modeling are included in digital format in the online data repository of this paper (Tables 1, 2, 3 and 4, and Supplementary Tables S1 to S6).

**Acknowledgement**

We would like to thank the captains, crews, and shipboard scientific parties of the R/V *Yokosuka* and the operating team of the submersible *Shinkai* 6500 for their great work during the YK16-01, YK18-08, and YK19-05S cruises. We used the submersible photos, rock samples, and survey information for these cruises provided in the Data and Sample Research System for Whole Cruise Information by JAMSTEC (http://www.godac.jamstec.go.jp/darwin/). The Kyoto University Research Reactor Institute is gratefully acknowledged in their assistance of undertaking the radiometric dating. We would like to express our great appreciation to Prof. T. Tsujimori (ORCiD: 0000-0001-9202-7312) for his effort in management of the laboratory at Tohoku University. We also thank R. Fukushima (ORCiD: 0000-0003-2683-6757) for improving the wording in the manuscript. We are really grateful Y. Matamura, Y. Shimbo, and Y. Jindo for their help and discussion on scientific matters. The authors would like to thank Enago (www.enago.jp) for the English language review. This manuscript was reviewed and improved by two anonymous reviewers, Topic Editor Coltorti, M., Executive Editor Muro, A.D., and the Editorial Board of Solid Earth. This research was supported by the Cooperative Program (No. 106, 202) of Atmosphere and Ocean Research Institute, The University of Tokyo. The Japan Society for the Promotion of Science (Grant Numbers 17K05715, 18H03733, 20K04098) also supported this research.

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
