# Peer review of "Contribution of carbonatite and recycled oceanic crust to petit-spot lavas on the western Pacific Plate"

_EGUsphere, 2023_

## Author Comment (AC1)

Reply to Referee Comment #1 (RC1)

Thank you for your remarks, comments, and suggested revision. We appreciate your time and effort in reviewing our manuscript.

We have considered the comments and revised accordingly. We have made changes to address the majority of the issues raised by the reviewer.

**Specific comments**

- Title: the title is totally un-representative of the content of the paper. A better title could be "Crustal recycled and carbonate components required for the genesis of petit-spot alkaline basalts", or something similar.

- We revised the title as "**Petit-spot lavas on the western Pacific Plate: contribution of carbonatite and recycled oceanic crust**"

- Abstract: the data in the manuscript do not contribute to define the origin nor the structure of the LAB. The model provided is only petrogenetic. The opening of the abstract must be modified.

- We revised the opening of the abstract:

"The lithosphere….partial melts. The petit-spot…the LAB (lines 44 to 48 in previous ver.)"

to

"Petit-spot volcanism, which occurs owing to the plate flexure, have been reported from around the world. As the petit-spot melts ascent from the asthenosphere, they provide the essential information of the lithosphere–asthenosphere boundary (LAB)."

Please see the lines 43 to 45 in the revised manuscript.

- Lines 74-82: The list of alkali basalts from random worldwide localities is not very meaningful, as readers might be not aware of each specific geodynamic context. Better re-shape this section, discriminating between tectonic settings and inferred geodynamic context, i.e. plume vs. not plume-related origin, continental vs. oceanic, etc….as it is done for petit-spot volcanoes shortly below. Intracontinental alkali basalts are not occurring only in China and North America; intraoceanic plume-related alkali basalts are not occurring only in Hawaii

- Thank you for your remarks. We revised the first sentence of introduction as

"Among the upper mantle-derived alkali basaltic lavas in oceanic settings, those on thicker plates away from the mid-ocean ridge, could be divided into plume-related and non-plume-related volcanoes. For example, …in Hawaii and Samoa (…. Hart et al., 2004; Konter and Jackson, 2012; Koppers et al., 2008; Reinhard et al., 2019; ….Yang et al., 2003). Non-plume related … magma (Hirano et al., 2006…Yamamoto et al., 2014, 2018, 2020)." Therefore, the occurrence of petit-spot volcanisms supports the presence of melt at lithosphere–asthenosphere boundary (LAB) below the area at least."

Please see the lines 72 to 82 in the revised manuscript.

- Lines 87-88: references are needed here! Falloon & Green 1989 EPSL; Falloon & Green 1990 Geology; Foley et al. 2009 Lithos; Ghosh et al. 2009 ChemGeo

- Thank you for your kind instruction. References were put in this sentence. Please see the lines 88 to 89 in the revised manuscript.

- Lines 124-131: In this (long) sentence, the concepts of LAB nature and regional petit-spot volcanism are put together, but I feel some explanation is missing. Petit-spot volcanism provides info about asthenospheric melting as many other types of volcanism do. Why should the reader find this type of magmatism particularly useful? This section is the key of the scientific background, as it links the "big picture" to the case study!

- Thank you for your kind remark. The usefulness of petit-spot volcanism is explained by their tectonic setting, that is, they erupted on seafloor without hotspot and ridge activity. We added the phrase in the end of sentence:

"Although there is…, the occurrence of petit-spot…the region because they erupted on the seafloor without hotspot and ridge activities (Hirano et al., 2006….)."

Please see the lines 140 to 143 in the revised manuscript.

- Section 4.4 (geochemical modelling) should be removed from the methodology and included later in the modelling/discussion section, at line 679 (maybe 697). At the moment is out of context, and forces the reader to go back to 4.4 section while being in the middle of the discussion about the nature of the mantle source and the partial melting processes.

- Section 4.4 was moved to Section 6.3. In addition, we deleted the non-modal batch melting model of simple garnet lherzolite. Please see the lines 693 to 716 in the revised manuscript.

- Geochronology: 40Ar/39Ar dating on these samples is for sure problematic, due to the intense state of alteration.

- As mentioned in the method section 4.3, we removed the alteration factors for the ArAr specimens. The descriptions about "alteration" in the section 6.2 are just about whole rocks. It is not about the ArAr specimens. Any secondary materials in the ArAr specimens are not recognized in the crystalline samples. Please see the newly described part in section 5.3 (line 503 to 510 in the revised manuscript).

However, at the moment the presentation of the data is not convincing enough to be acceptable for publication: no discussion about K/Ca as a function of alteration is provided

- Total K/Ca ratio of both samples is estimated using irradiated 39ArK/37ArCa ratio as newly shown in Table S4, which are mostly correspond to the molar K/Ca ratio before irradiation estimated using the bulk compositions. Please see the newly described part in section 5.3.

the negative age (-0.11 Ma) obtained for the sample R01 is simply meaningless, and obtained with only 49% of released 39Ar

- Yes, the value "-0.11 Ma" is meaningless and removed from the text as you designated the 49% released 39Ar might be insufficient. Description of the plateau age of the sample #1522 (right) is reduced in Fig.10a as well.

> why for sample 1466R6-001 was the inverse isochron age chosen?

- We adopted inverse isochron age of the sample 6K1466 R6-001 rather than plateau age because the initial 40/36 ratio is higher than atmospheric value (excess Ar) as mentioned in the section 5.3.

> More details are needed. On the other hand, the age ranges obtained using Fe-Mn crust or palagonite are interesting and useful.

- Please see the newly described part in section 5.3.

- Geochemistry: along the whole results and discussion sections, sample 1466-R6 and R7 are treated as "anomalous" Si-undersaturated rocks, more vesicular, ascribable to a genesis from a different magma suite and thus mantle source (lines 564). This is a crucial issue, because authors use this discrepancy to speculate about different mantle sources for the magmatism. However, from Table 2 it is clear that samples 1466-R6 and R7 have the highest LOI (up to 6 wt%), as reported by the authors themselves later on (lines 594-596; lines 604 "…alteration was not extensive…excluding sample 1466R7-001, 1466R7-003"). In Supplementary Fig. S1 we see that sample 1466-R7 contains the most Fe-rich olivine phenocrysts, but at the same time also altered peridotite xenoliths or xenocrysts are included in this sample (Fig. S4). The question is: is it possible that the whole-rock analyses are just effect of alteration, or have just been "doped" because xenoliths/xenocrysts fragments were simply mixed up with the host rock during sample preparation? This would also explain why Sr-Nd-Pb isotopes are not showing any difference with respect to the other samples (lines 575-576), not requiring any differential contribution of carbonatite flux to the source (lines 584-586).

- As you said, the 1466R6 and R7 basalts have some ultramafic xenoliths (~ 7 mm–10 mm) and xenocrysts. In fact, the majority of xenoliths are pyroxenite or pyroxene xenocryst (<5 mm; Mikuni et al., 2022PEPS) rather than peridotite. These pyroxenite xenocryst and pyroxenite xenoliths are composed of clinopyroxene with SiO2 of 48–

52 wt% and orthopyroxene with SiO2 of 51–54 wt%, and then, contamination of these xenoliths would not account for decreasing SiO2 content of 1466R6 and R7 basalt.

The variable composition of olivine in the 1466R6-001 might be due to the degree of hydration of them (i.e., Fe-rich olivines might be slightly iddingtisitized shown in Fig. 3a in previous manuscript), and we analyzed the olivine and clinopyroxene again following your later comment. Please see the new Table S1 and Fig. S1.

As large portions of fresh glass are analysable in these samples (Fig. 4a, BSE image), is it possible to compare the whole-rock analysis of sample 1466-R7 with the composition of its glass (as done for the other samples) and thus verify the difference between the two magmatic suites?

- Fresh glass was not observed in the 1466R6, R7 basalts, and then only whole-rock analysis was conducted for these samples.

- Chapter 6.2, containing the petrographic description of the samples and the evaluation of their alteration state and chemical representativeness, must be moved right at the beginning of the result section.

- We added the explanation of the lack of secondary phases in the petrography part of section 3.1, 3.2 and 3.3. The information of loss on ignition was moved in the section 5.1. Please see the lines 210-211, 234, 268, and 419-421 in the revised manuscript.

- Lines 670-673: Carbonatites are not depleted in U, Th and Nb. A typical feature is the high Nb/Ta, together with low Zr-Hf. Include reference, and/or plot an average carbonatite trace element pattern in Fig. 7 (using the same dataset as that for isotopes).

- Thank you for your correction. The sentence of lines 670 to 673 was modified and references were added as

"The depletions of specific elements (e.g., Ta, Zr, Hf, and Ti) of …mantle (Bizimis et al., 2003;…Hoernle et al., 2002; …)." Please see the lines 667, 669, and 670 in the revised manuscript.

- Why for major elements and isotopes are author using the whole carbonatites GEOROC database, but for the partial melting model (section 6.4 and Fig. 11) only the "average carbonatite" of Bizimis et al. (2003) is used?

- For the partial modelling of trace elements, we used the "average carbonatite" of Bizimis et al. (2003) because the trace element composition of carbonatite is very diverse as well as major and isotopic compositions. Bizimis et al. (2003) said that their "average carbonatite" is useful for geochemical modeling of trace element composition. In addition, we tried other carbonatite values which were commonly used in geochemical modeling as well (Hoernle et al., 2002), but they were not suitable to our petit-spot basalts and "average carbonatite" was fit to petti-spot.

- The "average carbonatite" of Bizimis et al. (2003) does not include the radiogenic isotopic compositions, then, in order to show the HIMU-EM1 trend of global carbonatites, we plot the whole carbonatites from GEOROC database in Sr-Nd-Pb isotopic plots.

We put the explanation about it. Please see the lines 716 to 719 in the revised manuscript.

- Lines 698-707: I agree the partial melting of garnet lherzolite with small crustal component + low carbonatite flux is the most plausible among the model performed, but still this model is not able to fully reproduce the Ta, La, Sm-Eu-Ti and Y concentrations. Is there a reason? Did authors try to change the modal composition of the source, melting contribution percentages and/or amount of recycled oceanic crustal material to better fit the petit-spot pattern?

- Thank you for your suggestion. We added the models using the smaller garnet modal ratio of the source, and Sm-Eu-Ti fit the petit-spot. For Y, the partition coefficient of White (2013) was not appropriate, so we changed the data to

McKenzie and O'Nations (1995), which fit well. Ta and La did not perfectly fit, but this may be a limit of such model with the assumption of carbonatite composition.

In addition to that, based on the comments of Reviewer #2, We added the model using the different melt connectivity threshold ($\alpha_c$). The explanation about that was also added.

Please see the new Fig. 11 with its caption (line 798-809), and line 694-696. 708-709, 713, 719-737.

- Lines 732-747, Isotopic modelling: as authors in the previous section model the percentage of recycled crustal components, peridotitic fertile mantle and carbonatitic flux, it would be extremely interesting to verify if such percentages are also applicable to the isotopic mixing models. Day and Hilton (2011, EPSL; 2021, Geology) and Day et al. (2022, GCA) tried to perform such model for the Canary Island magmatism; the same was done by Markse et al. (2008, JPet) and Pietruszka et al. (2013, EPSL) for Hawaii.

- Thank you for your suggestion. We checked the papers you kindly introduced us.

It would be interesting if the percentages in the trace element model could be applied to isotopes as well, but it is difficult to determine representative isotopic values for carbonatite more than trace element compositions. It is also difficult to assume that "carbonatite is HIMU(or EM1)", so the discussion here is limited to isotopic "trends" for petit-spots and carbonatites.

- Chapter 6.4 is entirely speculative, does not add value nor constraints to the model. I suggest to remove it up to line 785. The content of lines 785-792, that is the hypotheses made to reconcile the model with the geodynamic context of the area, can be incorporated at the end of section 6.3.

- Thank you for your suggestion. Considering the context at the end of Section 6-3 and comment of reviewer 2#, we included line 775 to 791 at the end of Section6-3 and a few phrase were modified.

 Please see the lines 772 to 796 in the revised manuscript.

- Lines 811-812: remove this sentence. In this paper the origin of LAB is not discussed at all.

- The sentence was deleted.

**Technical corrections**

- Line 73-74: the opening sentence is too broad, as it is having a poor meaning. Maybe correct to: "the petrogenesis of alkali basalts and the nature/evolution of their source mantle…". Which tectonic settings? Be more specific.

- According to your comments above (about lines 74 to 82), we modified this part. Please see the new introduction part in the revised manuscript.

- Lines 104-105: Maybe better re-phrasing to: "…to understand the nature of the underlying lithosphere-asthenosphere system and model the geodynamic evolution of the region"

- Thank you for your suggestion. We rephrased this part as your suggestion. Please see the line 109 to 110 in the revised manuscript.

- Lines 109-112: Maybe the concept can be smoothened and made more concise: "In the last 20 years, the increasing knowledge of petit-spot volcanic settings has provided useful insights on the nature of the lithosphere-asthenosphere system, especially in the NW Pacific region (Hirano et al., 2006, Hirano & Machida 2022)"

- Thank you for your suggestion. We modified here as your suggestion. Please see the line 114 to 115 in the revised manuscript.

- Lines 117-121: Sentence too long, can be cut to clarify the concept

- We separated the sentence into two sentences as

"Petit-spot melts, which originated from the asthenosphere unrelated to mantle plume, could be a key to elucidating the nature of the LAB (Hirano and Machida, 2022). Their asthenospheric origin was supported by MORB-like noble gas isotopic

ratios, multi-phase saturation experiment, and geochemistry (Hirano et al., 2006; Hirano and Machida, 2022; Machida et al., 2015, 2017; Yamamoto et al., 2018)."

Please see the line 121 to 125 in the revised manuscript.

- Lines 121-122: Where? Provide some names of localities, as this might be of broad interest to the reader!

- Thank you for your kind comments. We modified here as

"Recently, ...... worldwide including Java Trench, Tonga Trench, Chile Trench, Mariana Trench, Costa Rica, North American Basin and Range, and southern offshore of Greenland, implying the……magmatism (Axen et al., 2018; …)."

Please see the line 130 to 131 in the revised manuscript.

- Line 129: what does "hybrid factor" mean?

- Thank you for your remark. The phrase of "hybrid factor" and references were unsuitable.

The phrase of "…, or hybrid factor" was removed, and Audhkhasi and Singh (2022) and Herath et al. (2022) moved to the partial melting group (reference of "presence of partial melting").

Herath et al. said that the LAB beneath offshore New Zealand were structured. They suggested that the LAB was composed of three distinct layers including radial anisotropic layer, azimuthally anisotropic layer, and melt-rich channel.

We also add and fixed references: Kang and Karato (2023) was added at reference of "the physical property of minerals" group, Hua et al. (2023) was added at reference of "presence of partial melting" group, and Katsura and Fei (2020) was fixed as Katsura and Fei (2021).

Please see the line 136 to 139 in the revised manuscript.

- Line 276: EPMA analyses were not made on glasses only. Please correct and distinguish in this section the analytical conditions for glasses (defocussed

beam?) from those used for minerals. How about matrix correction (Pouchou and Pichoir 1991)?

- Sorry for lack of information. Analytical condition of mineral was added (beam diameter of glass: 10µm, mineral: 2µm), and some part were modified as

"A peak counting time of 20 s and a background counting time of 10 s were used, except for Ni, for which a peak counting time of 30 s and a background counting time of 15 s. For Na analysis of glass, the peak counting time was 5 s and the background counting time was 2 s."

Raw data were corrected using a ZAF online correction program, rather than PAP method. The description about it was also added.

Please see the method section in the revised manuscript.

- Line 417 and figure 4a: alkaline and sub-alkaline; dividing line is from Irvine and Baragar (1971)

- Thank you. We added the reference in the caption of Fig. 4a. Please see the caption of Fig 4 in the revised manuscript.

- Line 433: "…compared to the representative ocean island basalt (OIB)"

- This sentence was corrected as your comment. Please see the line 424 in the revised manuscript.

- Data from worldwide carbonatites presented in Fig. 9 are extremely scattered and not very useful, as they cover the entire diagram. Are they needed? If so, I recommend the authors to filter the dataset at least for the geodynamic settings or sample freshness/data quality.

- Thank you for your suggestion. We deleted the carbonatite data with 87Sr/86Sr > 0.706 and added the description. Please see the Fig. 9 in the revised manuscript.

- Line 556: "petrography and geochemistry".

- This sentence was corrected as your comment. Please see the line 559 in the revised manuscript.

- Line 607-608: not entirely true. Some samples are indeed altered, and are specifically those used to argue for the heterogeneity of the mantle!

- Thank you for your remark. The phrase was revised as:

  "These observations showed that most of the petit-spot basalts were unaffected by seawater alteration with a few exceptions."

Please see the line 607 to 608 in the revised manuscript. Please see reply to comment no. 8 for the major element composition of the 6K#1466R7-001 and R7-003 basalts.

- Lines 617-619: "mass balance calculation of fractional phases….could not be performed because of inadequate phenocrysts" what does it mean?

- Thank you for your question. The phrase was very unclear, sorry.

  As described in lines 613–617, petit-spot basalts have no phenocryst (or have only micro-phenocryst), and petit-spot magma might have experienced crystal fractionation during its magma-stagnation within the lithosphere. Therefore, the calculation of primary composition of basalts using the mineral mode on the thin section cannot be conducted. The phrase was revised. Please see the line 613 to 618 in the revised manuscript.

- Lines 668-670: concept repeated.

- Thank you for your remark. This sentence was deleted.

- Line 729: Herzberg

  - Thank you for your correction. Please see the line 752 in the revised manuscript.

  **Figures and Tables**

- 1 can be re-organized in a 2x2 panel, as at the moment the panels are too small to be read properly. In the tomographic image to the right, maybe it would be useful to have also a more "zoomed" view of the upper 700-800 km.

- Thank you for your instruction. Figure 1 was changed to 2x2panel. Considering the composition of the 4 panels and the available space, we prepared the tomographic image as a vector file rather than cutting out an upper mantle portion, so that it can be clearly seen when enlarged (but it is pasted as an image file in the submitted file). Please see the new Fig. 1.

- Line 144: Replace "although" with "Notwithstanding"

  - Thank you for your correction. Please see the line 156 in the revised manuscript.

- 3: mineral labels should be added to all the optical microscope photos and BSE images. Why not splitting the figure in 4 different figures, or move some material to the supplementary information? As it is, it cannot be presented as a single figure composed by 4 multi-panels…

- Thank you for your remarks. Mineral labels were added to all photomicrographs and BSE images. We think these figures can be vertically organized as Fig. 6 of Mikuni et al. (2022 *PEPS*) https://doi.org/10.1186/s40645-022-00518-y. If it is difficult, we will try another way.

- 4: Panel b does not look very helpful, could be replaced with a K2O vs Na2O diagram to discriminate Na- from K-affinity. Also, plotting the literature data for kimberlites causes a "squeezing" of the OIB data down to K2O/Na2O <1.5, with all the data overlapping with each other. Is this really necessary?

- Thank you for your suggestion. Fig. 4b was changed to $K_2O$ vs $Na_2O$ diagram, and data of kimberlite were deleted. OIB and MORB data from PetDB were changed to those from "Expert datasets" of Stracke et al. (2022) same as the supplemental Figure S3 and S7.

- 5-6: I would recommend to plot MgO wt% in reverse order on the x-axis, so that to increase the degree of differentiation rightwards. This would also mark better the differences with respect to NW Pacific rocks, clearly visible on the CaO/Al2O3 vs. MgO plot.

- Thank you for your suggestion. We revised Fig.5 and Fig.6 to plot MgO wt% in reverse order on the x-axis as you said. Please see the new Fig. 5 and Fig. 6.

- 7 caption: "Primitive-mantle- (PM, Sun & McDonough, 1989) normalized trace element…..". Delete the last sentence.

 - Thank you for your correction. Please see the caption of new Fig. 7 in the revised manuscript.

- 9: carbonatite symbols in light green are poorly readable. The details about the mixing model in the caption can be moved to the supplementary material.

- Thank you for your remark. We changed the symbol color of carbonatite to dark blue. Please see the new Fig. 9.

- 10b: provide the reference for the age of the detrital zircon in the Fig. caption.

 - Thank you for your remark. Please see the caption of Fig. 10 in the revised manuscript.

- S9. Is it different from Fig. 11 in the main text? If not, why duplicate it?

- This figure is included in the supplement because the model for these samples (1203, 1206, 1542, 1544) was based on "if the trace element compositions of these samples were primary". Please see the line 732 to 735 in the revised manuscript.

- Table 2: Is this dataset including both glass analyses performed by EPMA and LA-ICP-MS and whole-rock analyses performed by XRF? If so, it must be clearly defined, and possibly the 2 samples set must be distinguishable from each other, not mixed. Values of 0.00 have no sense. Are they not analysed (n.a.) or below detection limit (b.d.l.)?

- Thank you for your instruction. 0.00 was converted to "-", and the explanation was described below each table ("-": not detected or <0.001 for cation numbers).

  The analytical method was described in each table, and table of major and trace element data were divided into Table 2 and 3.

- Supplementary Table S1: mineral chemistry analyses must be filtered for data quality, at the moment they are totally un-presentable. Some olivine analyses sum up to <98 or >101 (sometimes >102!!!!), must be discarded. Some olivine

analyses have Al2O3 >0.2 wt% (sometimes >1 wt%): must be deleted, are just mixed analyses. Filters must also be applied according to the quality of apfu calculations. Values 0.00 are not presentable: are they be not analysed (n.a.) or below detection limit (b.d.l.)? The same applies for cpx and plag, as many grains sum up to <98. Many spinel analyses have >0.8 wt% SiO2, must be deleted. One spinel analysis has 15 wt% SiO2, it is just a mixed analysis, delete.

- Thank you for your remark. The compositions of minerals were analyzed again using EPMA in GSJ, AIST. We selected olivine, clinopyroxene, and plagioclase (only 1466R3 basalt). The data of FeTi oxide and spinel would not be presented because they are not significant in discussion of the study.

- Please see the method section (line 290-298), Fig. S1, Tables S1, S2 and S3 in the revised manuscript. For clinopyroxene, we present the Mg# vs. TiO2 and ternary diagram, and the Al4 vs Al6 plot was deleted due to lack of sense in discussion. We also added the description of another EPMA (JXA-iHP200F) for mineral re-analysis in the Section 4.1.

**The revised points other than the reviewer's pointed out.**

・We verified that the glass analysis data obtained through EPMA represented an average of 10 points. Subsequently, this information was incorporated into the method section. Additionally, Table 2 includes 2-sigma values, and the notation "n=10" has been included.

・The term "detrital zircon" was amended to "zircon in peperite."

・Certain passages were changed from past tense to present tense.

・Symbols representing NW Pacific petit-spots were standardized.

・The PM-normalized trace element range of NW Pacific petit-spots (Fig. 7g) was adjusted using data from Hirano and Machida (2022).

---

## Author Comment (AC2)

Reply to Referee Comment #2 (RC2)

Thank you for your feedback, comment, and suggested revision. We appreciate your time and effort in reviewing our manuscript.

We have considered the comments and taken action accordingly. We have made changes to address the majority of the issues raised by the reviewer.

- First, the geochemical model provided is unsufficiently explained. Although the model used refers to another study from Ozawa et al. (2001), some aspects regarding the different parameters used and how their variations could impact the final results would deserve to be more explained and explored. As an example, this is stated in the manuscript that flux of small amounts of carbonatitic melts are involved in forming petit-spot lavas, but the notion of small is relative. Could it be possible to have a more quantitative estimation of such amounts in %, or in terms of carbon content in the lherzolite source even approximative? Besides, could we consider drastic changes in the ultimate model results by varying the critical melt fraction of a certain amount? I suppose the latter could vary depending e.g. on the melt composition ('better' connectivity of a carbonatitic liquid compared to a basalt typically).

- Thank you for your comment. This is mass-balance based model, and then a flux ratio of 0.1 means that there is a 10% flux contribution to the mass of the melting source. In other words, $\gamma = 0.1$ means a flux of 0.1 for a melting source of mass 1. Therefore, we have rephrased the "low carbonatite influx" to "10% carbonatite influx to a given mass of source".

We also conduced the model considering the different garnet mode and melt connectivity (critical melt fraction) in addition to previously conducted models. Please see the new Fig. 11, its caption, and line 708-708, 717-737 and 827-828 in the revised manuscript.

- Regarding the origin of fluxing carbonatitic melts as discussed in l. 775-791 in section 6.4, some sources are discussed: "subducted carbonated pelite, pyroxenite/eclogite, or peridotite stored as diamond or metal carbide in the reduced lower portion of the upper mantle(….or) the existence of a carbonate-rich layer in the LAB owing to the horizontally spread carbonate from around the wedge mantle rather than upwelling from the deep mantle". Carbonatites can be produced by melting of such lithologies, and I do not contest these possible origins, but since decades now, plethora of publications in experimental and modeling petrology have demonstrated upper parts of the asthenosphere, where the bottom layer of LAB is probably located, can be enough oxidized to have an 'original' source of carbon hosted as carbonates, and within the P-T conditions prevailing in this region, such carbonates can melt to form carbonatites (see e.g. Massuyeau et al., 2021). In this convective mantle, deep we get carbonatites and by ascending into the mantle, the melt composition gets close to a basalt with a kind of opposite trend between the enrichment in silica and the carbon depletion. Besides carbonatites can also be stabilized in the deepest and cold parts of the lithosphere. So, this scenario appears as most likely to explain formation of carbonatites in the LAB region than those explained above as source for carbonatites, and should at least deserve to be discussed too.

- Thank you for your comments and instructions. We checked Massuyeau et al. (2021) along with Massuyeau et al. (2015).

  The introduced papers are very interesting, and we added the discussion based on them in the introduction part and discussion part. We newly described another origin of carbonatite and depth-dependent chemical variation; carbonatite, kimberlite, and alkali basalt. Please see the line 89-94, 787-793 in the revised manuscript.

- 44-46 : The two first sentences of the abstract should be reworked. As it, it can let imagine the manuscript will discuss about the origin of the geophysical discontinuities observed at LAB, while this is not the purpose of the manuscript. And the first sentence could be removed to my mind.

- Thank you for your comments. Another reviewer also pointed out this part as:

Reviewer#1 " the data in the manuscript do not contribute to define the origin nor the structure of the LAB. The model provided is only petrogenetic. The opening of the abstract must be modified."

We revised here as:

"Petit-spot volcanism, which occurs owing to the plate flexure, were reported around the world. The petit-spot melts ascent from the asthenosphere, and provide the essential information of the upper mantle."

Please see the line 43 to 45 in the revised manuscript.

- 57-58 : "… *partial melting of garnet lherzolite with a small degree of carbonatite melt flux with 58 crustal components.*" à I understand the crustal component are brought by carbonatitic melts, which is probably not the case here, right?

- Yes. In our model, carbonatite is the influx and crustal component is mixed in the source. However, both contribute to sources of melting.

- 59-60 : "…*and provides an implication for the genesis of tectonic-induced volcanism with similar geochemical signatures to those of petit-spots.*" à I do not understand this sentence.

- This meant that tectonically induced volcanoes with trace element compositions similar to our sample, such as NorthArch and other petit-spit like rejuvenated volcanoes (such as Tonga Trench), could also be explained by a similar model.

We revised the phrase as:

"… and provides an implication for the genesis of tectonic-induced volcanoes including Hawaiian North Arch volcanics and Samoan petit-spot-like rejuvenated volcanoes having similar trace element composition to petit-spot basalts."

Please see the line 57 to 58 in the revised manuscript.

- 68 : add a reference to the geochemical modeling done in the study

- Thank you for suggestion. We added the phrase about it:

" ….LAB. We conducted geochemistry, geochronology, and geochemical modeling for petit-spot volcanoes on the…."

Please see the line 66 to 67 in the revised manuscript.

- 88-89 : It would be interesting to mention that with variations of P-T, as it can occur at LAB potentially, melting of carbonated peridotite can also produce a chemical continuum in the composition of produced melts, from carbonatite to basalt, with intermediate terms like kimberlites, nephelinite, melilitite, etc. It could possibly partly explain the variations obtained in terms of major elements between different series.

- Thank you for your advice. We revised this part as:

"In addition, carbonatites and Si-undersaturated melts are generated through partial melting of $CO_2$-bearing or carbonated peridotite. The produced melts could exhibit continuous chemical variations depending on pressure (i.e., depth), that is, carbonatitic melts are produced in the deep asthenosphere (300 km to 110 km), while carbonated or alkalic silicate melts are generated in the shallower upper mantle (~110 km to ~75 or 60 km) (Keshav and Gudfinnsson, 2013; Massuyeau et al., 2015, 2021)."

Please see the line 89 to 94 in the revised manuscript.

- 100 : Instead of "*carbonatitic materials*", I would rather use "*carbonated materials*".

- Thank you for your remark. We revised the phrase as you said. Please see the line 104 in the revised manuscript.

- 101-105 : As the notion of LAB and its relationships with both mantle melting and geophysical properties have not been discussed yet (coming next in the Background section), I would replace "LAB" by "the uppermost part of the asthenosphere" or similar, something more neutral in the description.

- Thank you for your instruction. We revised "the LAB" to "the uppermost part of the asthenosphere" as you mentioned. Please see the line 105 in the revised manuscript.

- 119 : As you mention the LAB, it would be appropriate to have a short and general discussion/presentation about the latter and its relationships with mantle melting and geophysical signatures. Besides, once considered the notion of LAB, could we consider the bottom of a possibly metasomatized lithosphere as a possible source of petit-spot lavas too ?

-Thank you for your suggestion. We added the sentence about LAB as:

"The LAB is identified as a discontinuous transition in seismic velocities at the base of the lithosphere, and its causes are attributed to hydration, melting, and mineral anisotropy with considerations for the unique characteristics in each tectonic setting (e.g., Rychert and Shearer, 2009). The occurrence of petit-spot volcanism substantiates the existence of melt at the LAB below the area at least (Hirano et al., 2006)"

Please see the line 125 to 129 in the revised manuscript.

The bottom of metasomatized lithosphere as a source of melts at LAB was discussed as a metasomatized amphibole-rich vein in the lithosphere (Section 6-3, line 681 to 688).

- 143 : The authors mention a low-velocity zone, but no reference is mentioned. Same in line 144-145. Moreover, it would deserve more explanation, why do they need to mention. This is kind of information from nowhere.

- Thank you for your remark. We added some description of vicinity seamounts with references, the reference of seismic data, and that no heat supplies have been reported. Please see the line 156 to 159 in the revised manuscript.

- 451-452 : Here authors should also precise that the samples 1203 and 1206 are from another study (Hirano et al., 2019)

- Thank you for your remark. We added the description and reference in the caption of Fig.4 (1203 and 1206; Hirano et al., 2019 and 1466R7; Mikuni et al., 2022). Please see the caption of Fig. 4 in the revised manuscript.

- 452-454 : Any kind of filtering done from the PetDB database ?

- The plotted data were not filtered, and then we compiled the data again using "Expert data set of Stracke et al. (2022)" for MORB and OIB which can be clearly seen the information of data. Additionally, following the comment of reviewer#1 of "*Plotting the literature data for kimberlites causes a "squeezing" of the OIB data down to K2O/Na2O <1.5, with all the data overlapping with each other. Is this really necessary?*", we changed the Fig. 4b to Na2O vs. K2O plot, and kimberlite plot were discarded. The max K2O/Na2O value from PetDB was only used in the Figure.

Please see the new Fig. 4 and its caption.

- 488 : "*with Bizimis et al. (2003)*" à it could be of interest to precise this point, e.g. as part of the Supplementary part

  - Thank you for your suggestion. The symbols of the carbonatite by Bizimis et al. (2003) were highlighted with a border in Fig. 9, and they show the clear HIMU-EM1 trend.

- 564 : "*were suggested*" à By who ? If this is the authors suggestion, they should use the present tense then, here readers could think they are mentioning another reference not cited.

- Thank you for your instruction. The phrase of "were suggested to" was changed to "are implied to". Please see the line 567 in the revised manuscript.

- 595-596 : Could we simply imagine this LOI being a volatile (CO2-H2O) rest of the primary melt ? Does the Figure 8 completely rejects this hypothesis ? In terms of major elements (Fig. 5), it seems difficult to exclude completely this hypothesis relatively to other samples, except maybe the high FeO content. Does the latter could maybe favor your alteration scenario ?

655-660 : To reconsider in light of the previous comment regarding the nature of 1466R7.

- Certainly, among the two samples characterized as altered rocks, namely 1466R7-001 and R7-003, it might be reasonable to consider R7-001 with an LOI of 2.86 as a more primary sample, as suggested by the Sm/Hf ratio vs. MgO plot. However, it seems unlikely that the LOI is indicative of primary $CO_2$ or $H_2O$. This is because, according to Okumura and Hirano (2013 geology), the post-eruption $CO_2$ content in NW Pacific petit-spot samples, even at its highest, is 1200 ppm, with $H_2O$ content at 0.6 wt%. Furthermore, the absence of minerals such as amphibole or mica, unlike in kimberlites, suggests that these volatile components are of secondary origin.

- The absence of fresh glass, the prevalence of hydrated olivine, and some spikes in trace elements like negative anomalies in Sr further led us to refrain from designating it as a representative sample.

• 741-744 : It seems to be in contradiction with what is said in previous sentences, no ?.. Maybe this sentence would need to be reworked a bit.

- We wanted to insist that we cannot connect the mantle endmember (HIMU and EM-1) to carbonatite or recycled crust, respectively, and it is difficult to suppose representative isotopic ratios of them (particularly carbonatite).

We revised here as:

"The variability of global carbonatite isotopic compositions also makes it difficult to determine their representative isotope ratios (Fig. 9). Although such issues make a quantitative isotopic mixing model challenging, the HIMU-EM-1 like trend of the global petit-spot volcanoes may reflect the involvement of carbonatitic and recycled crustal materials."

Please see the line 766 to 769 in the revised manuscript.

• 767-770 : I do not understand the point here, it seems both sentences repeat the same point with different locations for 'representative' carbonatites.

- Thank you for your remark. As the reviewer #1 suggested to remove most of Section 6-4, we deleted the line 765 to 774 in the firstly submitted manuscript including line 767−770.

- 780-781 : "transition oxidation state" à it would deserve more explanation

  - Thank you for your remark. We added the phrase as (under lined part):

  "Subducted carbonated pelite, for example, would melt under high pressure (>8 GPa) through the oxidation at the redox boundary where the the iron-wüstite (IW) buffer changes to the quartz–fayalite–magnetite (QFM) buffer (i.e., redox melting; Grassi and Schmidt, 2011)."

  Please see the line 778 to 780 in the revised manuscript.

  **Technical corrections**

  Thank you for your careful review and taking the time out of your busy schedule. We revised as your corrections. For other cases, some comments are included.

- 52 : remove "*their*"

- 65 : replace "*moving*" with "*motion*" (line 64)

- 65 : replace "*on*" with "*over*" (line 64)

- 92 : "*to be explained*" (line 97)

- 101 : replace "*LAB*" with "*Lithosphere-Asthenosphere Boundary (LAB)*"

- Following your previous comment, we changed the phase LAB to "the uppermost part of the asthenosphere" (line 105)

- 119-120 : replace "*leading to an understanding*" with "*and further leading to a better understanding*"

- Following reviewer#1's comment, we changed the sentence here. (line 121 to 125)

- 144 : replace "*Although the*" with "*In spite of*"

- Following reviewer#1's comment, we replace here with "Notwithstanding". (line 156)

- 219 : "*and comprised*" (line 233)

- 329 : replace "*The*" with "*Both*" (line 346)

- 344 : replace "*value was*" with "*values were*" (line 361)

- 377-378 : replace "*In the model in this study*" with "*In this present study, the model uses a critical melt fraction*", and remove "*was*" in l. 379. (line 694 to 695)

- 544 : replace "was" with "is", the petit-spot volcanic field still exists I guess ;-) (line 546)

- 566 : "*K#1466R3*" right ?

- That's right. Thank you. (line 569)

- 570 : replace "*this*" with "*these*" we revised "this study samples" to "these samples". (line 573)

- 590 : replace "might affect" with "*might have affected*" (line 593)

- 733 : remove one "*the*"

- 787 : "*Hammouda et al. 2021*" ?

Sorry, Hammouda et al. (2020) is correct, and the caption of Fig. 12 was incorrect. We revised the reference in the caption of Fig. 12.

**The revised points other than the reviewer's pointed out.**

・We verified that the glass analysis data obtained through EPMA represented an average of 10 points. Subsequently, this information was incorporated into the method section. Additionally, Table 2 includes 2-sigma values, and the notation "n=10" has been included.

・The term "detrital zircon" was amended to "zircon in peperite."

・Certain passages were changed from past tense to present tense.

・Symbols representing NW Pacific petit-spots were standardized.

・The PM-normalized trace element range of NW Pacific petit-spots (Fig. 7g) was adjusted using data from Hirano and Machida (2022).

---

## Author Response (AR2)

Reply to Editor Comment

Dear Dr. Massimo Coltorti and Editorial support of *Solid Earth*

Thank you for your comments and suggested minor revision. We appreciate your time and effort in reviewing our manuscript.

We have considered the comments and revised accordingly, and have made changes to address the majority of the issues.

The figures applied to the Color Blindness Simulator, and were entirely created by the authors.

The figures in the manuscript appear to be low resolution due to being compiled into a single Word file as an Image file, but each figure is actually prepared as a Vector file. Please excuse any difficulty in readability.

**Reply to comments (grey: review comment, black: reply)**

・I consider your answer to their comments appropriate and pretty exhaustive. The case study is very interesting and your modelling rather robust, providing information and models useful to the scientific community.
Before it goes public however I also had some comments and suggestions to submit to your attention.

- Thank you for your kind comments and effort in reviewing our manuscript.

・As first I would suggest lightening the weight of your belief all along the text, making the role of the good data you have less important. As an example you state right at the beginning that melt is present at the LAB, giving your idea on the nature of this discontinuity. But this means that melt is physically already present or may be derived from the asthenosphere upwelling due to the flexure of the plate or mantle can melt in response to carbonatite influx? From my scientific (and philosophical) point of view it is nicer to first consider the various hypotheses and at the end explain why you favour one, that is yours.

- The origin of melts at LAB is indeed an important issue. Solving the problem from the chemical composition of our rocks is a challenging. Studying petit-spot volcanism, we have ever been grappling with this problem. The role of our model lies in demonstrating the definitive contribution of carbonatite to LAB melt. Given the understanding derived

from years of research that carbon-rich components can lower the mantle solidus, we believe it is certain that carbon-rich components play a role in the generation of melts at LAB.

- Based on your comment, lines 83 to 87 (in previous manuscript) in the *introduction* was modified as follows:

From (before the revisions)

"The presence of melt in the uppermost asthenosphere could be due to small-scale convection, heating, or the presence of hydrous or carbonatitic components (Hua et al., 2023; Korenaga, 2020). In particular, the presence of $CO_2$ and carbonated/carbonatitic materials is key in the formation of alkaline, silica-undersaturated melt in the upper mantle (Dasgupta and Hirschmann, 2006; Dasgupta et al., 2007, 2013; Kiseeva et al., 2013; Novella et al., 2014). ……."

To (after the revisions)

"The occurrence of melt in the uppermost asthenosphere could be attributed to small-scale convection, the presence of hydrous or carbonatitic component, or the uplift of the lithosphere in response to plate flexure; however, the possibility of such an occurrence remains ambiguous (e.g., Bianco et al., 2005; Hua et al., 2023; Korenaga, 2020). The presence of CO2 and carbonated/carbonatitic materials is a significant factor in the formation of alkaline, silica-undersaturated melt in the upper mantle (Dasgupta and Hirschmann, 2006; Dasgupta et al., 2007, 2013; Kiseeva et al., 2013; Novella et al., 2014). ……"

Please see the line 83 to 89 in the revised manuscript. (highlighted as yellow are modified part)

By having this sentence, it connects to the last sentence in the discussion part (Sect 6-3), which reads,

"Although the multiple origins of carbonatite are merely suggested and remain unclear, carbon-rich components play a key role in the partial melting of mantle at the LAB (Sifré et al., 2014), constituting the source of petit-spot magma." (line 803-805 in the revised manuscript)

Thank you for your suggestion.

・I found appropriate the change of the title, as suggested by the first reviewer, and with respect to the new title I will go a bit further, changing the order of the two sentences making the tiel even more focussed, that is "Contribution of carbonatite and recycled oceanic crust to petit-spot lavas on the western Pacific Plate".

-Thank you for your suggestion. We revised the title on your (and reviewer#1's) comments as follows: "Contribution of carbonatite and recycled oceanic crust to petit-spot lavas on the western Pacific Plate".
Please see the new title of our revised manuscript.

・I also suggest homogenising the labelling of the samples and always using the same label along the text. At the beginning of chapter 5 you say that dive 6K#1521 samples was labelled 1521 samples or 1466 for dive 6K#1466, but after I see various labels in the text, such as 6K#1466R3-001 and R3-004, which become 1466 R3 in the figures, or simply R3-001 or R3 basalts or 6K#1466R3 series, and so on… In this way it is going to be quite difficult for the reader (at least it was for me) to understand which is which.

- Thank you for your kind remarks. In the Section 5 and beyond, we have removed the notation "6K#" from the descriptions in the samples and unified the expressions like "1466R3 basalts, 1522 basalts, etc.".
- In the case of listing information about submersible dives, we have standardized the expression to "6K#1542 and #1544 dives".
  Please see the revised manuscript and highlighted (tracked) parts in the tracking file.

・Part of your modelling needs to be more circumstantial. For example, put directly in the text how much olivine you are adding to your least differentiated basalts, avoiding for the reader to jump in the supplementary and decipher it from the figure and table (row 755 →606?).

- Thank you for your comments.
  We rephrased the description of olivine maximum fractionation model in the line 660–662 (previous file) as:

"By applying the olivine maximum fractionation model (Takahashi et al., 1986; Tatsumi et al., 1983) to test two samples, it was noted that 7–9% olivine addition was required to achieve the olivine composition corresponding to "Mantle olivine array" in the NiO and Fo# spaces (Figs. S6a, b). The calculated primary trace element contents did not considerably differ from those of the analytical compositions (Table S5 and Fig. S6)." Please see the line 664 to 669 in the revised manuscript.

・ Or when you speak about different composition or degree of partial melting (row 660→755?), which of the two parameters you think is acting? If you can explain the compositional variations varying the degree of partial melting, you do not need two different sources. Or when your melting mode foresees negative, that is recrystallization of opx, if I understood correctly (row 806 →706?), which I found somehow peculiar in order to generate primary melt with 47% of silica.

- Thank you for your remarks. Unless the degree of partial melt is extremely large (>~10%), it does not significantly influence the model results.

The model results did not vary significantly due to the melting reaction (e.g. ol 0.03, opx 0.03, cpx 0.44, grt 0.500; Johnson et al., 1990).

[Figure]

Left: Melting mode of Johnson et al. (1990)          Right: This study (melting mode of Walter (1998).

All other parameters are the same.

"The calculated primary SiO2 for one of the 1522 basalts, obtained through olivine addition, was 45.8%. This may be slightly higher than that expected, but this is comparable to the case of alkaline lavas on the rejuvenated stage in Hawaii. The Hawaiian rejuvenated lavas modeled for trace elements during Opx crystallization reaction showed an average SiO2 content of 45.9 wt% and a maximum of 47.6 wt% (Borisova and Tilhac in 2021).

Considering the observed HIMU-EM-1 trend in radiogenic isotopes in global petit-spot basalts, our samples appear to have a smaller contribution of EM-1 components compared to NW Pacific petit-spot basalts. As mentioned in the main text, it is uncertain to definitively determine the specific contributions of each mantle endmember. However, the differences in isotopic ratios correspond to the variations of trace element patterns between NW Pacific petit-spots and our samples, which means that the trace element characteristics reflect the source. Here, we would like to emphasize the potential contribution of both carbonatite and the crust to the melting source of petit-spot lavas at least.

・Please consider also the opportunity to have the manus being read by a mother language person, that could improve the reading and the appreciation of your work.

- Thank you for your suggestion. The manuscript was revised by English Correction Service of "Enago" again.
  Please see the revised manuscript and highlighted (tracked) parts in the tracking file.

**Other revises**

・We added the reference (Wessel et al., 2019) about "GMT", the tool using making the bathymetric maps. Please see the captions of Figs. 1, 2 and 3, and reference list.
・All other corrections, except where noted in the peer review, were made in accordance with English Editing (Enago).

---

## Author Response (AR3)

Dear Editorial support of *Solid Earth*

Thank you very much for accepting the paper. We appreciate the diligent work of editors, reviewers, and editorial board. Below, we have listed a little modifications made in the final submission file:

**1. Figure 3 Caption and Acknowledgments:**
In Figure 3's caption and the acknowledgments section, we have included a statement regarding JAMSTEC's cruise database (DARWIN).

**2. Acknowledgments:**
We have added acknowledgments to the reviewers and editors. Additionally, we corrected a typographical error in the acknowledgment section, changing "greatfully" to "gratefully."

**3. Data Availability Section:**
We have revised the Data Availability section to reflect that we have decided not to store the data in EarthChem. The updated statement is as follows: "The data newly analyzed in this study and results of geochemical modeling are included in digital format in the online data repository of this paper (Tables 1, 2, 3, and 4, and Supplementary Tables S1 to S6)."

**Additional Notes:**

The tables in the "Text_Mikuni_et_al" file are in Excel format, accessible by double-clicking for comprehensive content viewing.
Supplementary Tables are included in a zip file in spreadsheet format. This file contains the tables mentioned in the main text as well.

Thank you for your continued support.

Best regards,

Kazuto Mikuni